# Impact and cost-effectiveness of the 6-month BPaLM regimen for rifampicin-resistant tuberculosis in Moldova: A mathematical modeling analysis

Lyndon P. James[1,2]*, Fayette Klaassen[3], Sedona Sweeney[4], Jennifer Furin[5], Molly F. Franke[5], Reza Yaesoubi[6], Dumitru Chesov[7,8], Nelly Ciobanu[9], Alexandru Codreanu[9], Valeriu Crudu[9], Ted Cohen[10], Nicolas A. Menzies[2,3]

1 PhD Program in Health Policy, Harvard University, Cambridge, Massachusetts, United States of America, 2 Center for Health Decision Science, Harvard T. H. Chan School of Public Health, Boston, Massachusetts, United States of America, 3 Department of Global Health and Population, Harvard T. H. Chan School of Public Health, Boston, Massachusetts, United States of America, 4 Faculty of Public Health and Policy, London School of Hygiene and Tropical Medicine, London, United Kingdom, 5 Department of Global Health and Social Medicine, Harvard Medical School, Boston, Massachusetts, United States of America, 6 Department of Health Policy and Management, Yale School of Public Health, New Haven, Connecticut, United States of America, 7 Discipline of Pneumology and Allergology, Nicolae Testemitanu State University of Medicine and Pharmacy, Chişinău, Moldova, 8 Clinical Infectious Diseases, Research Center Borstel, Borstel, Germany, 9 Chiril Draganiuc Institute of Phthisiopneumology, Chişinău, Moldova, 10 Department of Epidemiology and Microbial Diseases, Yale School of Public Health, New Haven, Connecticut, United States of America

* lyndonpjames@gmail.com

## Abstract

### Background

Emerging evidence suggests that shortened, simplified treatment regimens for rifampicin-resistant tuberculosis (RR-TB) can achieve comparable end-of-treatment (EOT) outcomes to longer regimens. We compared a 6-month regimen containing bedaquiline, pretomanid, linezolid, and moxifloxacin (BPaLM) to a standard of care strategy using a 9- or 18-month regimen depending on whether fluoroquinolone resistance (FQ-R) was detected on drug susceptibility testing (DST).

### Methods and findings

The primary objective was to determine whether 6 months of BPaLM is a cost-effective treatment strategy for RR-TB. We used genomic and demographic data to parameterize a mathematical model estimating long-term health outcomes measured in quality-adjusted life years (QALYs) and lifetime costs in 2022 USD ($) for each treatment strategy for patients 15 years and older diagnosed with pulmonary RR-TB in Moldova, a country with a high burden of TB drug resistance. For each individual, we simulated the natural history of TB and associated treatment outcomes, as well as the process of acquiring resistance to each of 12 anti-TB drugs. Compared to the standard of care, 6 months of BPaLM was cost-effective. This strategy was estimated to reduce lifetime costs by $3,366 (95% UI: [1,465, 5,742] *p* <

**Data Availability Statement:** All M. tuberculosis genomic sequencing data was from a publicly

available source at https://www.ncbi.nlm.nih.gov/bioproject/PRJNA736718 We applied our own exclusion criteria to that data and include this in the data repository, along with all other relevant data and code at https://github.com/lyndonpjames/BPaLM_Moldova/tree/main.

**Funding:** This publication was made possible by Grant Numbers T32 AI007433 (LPJ) and R01 AI146555-02 (NAM, TC) from the National Institute of Allergy and Infectious Diseases https://www.niaid.nih.gov/. The funding agency played no role in the study design, data collection and analysis, decision to publish, or preparation of the manuscript. The manuscript contents are solely the responsibility of the authors and do not necessarily represent the official views of the NIH.

**Competing interests:** JF has received grant funding from the Stop TB Partnership's Global Drug Facility to support the roll out of child-friendly formulations of second-line TB drugs.

**Abbreviations:** BPaL, bedaquiline, pretomanid, linezolid; BPaLC, bedaquiline, pretomanid, linezolid, clofazimine; BPaLM, bedaquiline, pretomanid, linezolid, moxifloxacin; CDF, cumulative distribution function; CI, confidence interval; CrI, credibility interval; CPI, Consumer Price Index; DST, drug susceptibility testing; FQ-R, fluoroquinolone-resistant; FQ-S, fluoroquinolone-susceptible; LJ, Lowenstein–Jensen; MDL, Moldovan Leu; GDP, Gross Domestic Product; GEL, Georgian Lei; HIV, human immunodeficiency virus; LTFU, lost to follow up; LY, life year; M. tb., Mycobacterium tuberculosis; MDR-TB, multidrug-resistant tuberculosis; NHB, net health benefit; QALY, quality-adjusted life year; RR-TB, rifampicin-resistant tuberculosis; SAE, grade 4–5 severe adverse event; SEM, standard error of the mean; SMR, standardized mortality ratio; TB, tuberculosis; UI, uncertainty interval; USD, United States dollars; WTP, willingness-to-pay; XDR-TB, extensively drug-resistant tuberculosis.

0.001) per individual, with a nonsignificant change in QALYs (−0.06; 95% UI: [−0.49, 0.03] $p$ = 0.790). For those stopping moxifloxacin under the BPaLM regimen, continuing with BPaL plus clofazimine (BPaLC) provided more QALYs at lower cost than continuing with BPaL alone. Strategies based on 6 months of BPaLM had at least a 93% chance of being cost-effective, so long as BPaLC was continued in the event of stopping moxifloxacin. BPaLM for 6 months also reduced the average time spent with TB resistant to amikacin, bedaquiline, clofazimine, cycloserine, moxifloxacin, and pyrazinamide, while it increased the average time spent with TB resistant to delamanid and pretomanid. Sensitivity analyses showed 6 months of BPaLM to be cost-effective across a broad range of values for the relative effectiveness of BPaLM, and the proportion of the cohort with FQ-R. Compared to the standard of care, 6 months of BPaLM would be expected to save Moldova's national TB program budget $7.1 million (95% UI: [1.3 million, 15.4 million] $p$ = 0.002) over the 5-year period from implementation. Our analysis did not account for all possible interactions between specific drugs with regard to treatment outcomes, resistance acquisition, or the consequences of specific types of severe adverse events, nor did we model how the intervention may affect TB transmission dynamics.

## Conclusions

Compared to standard of care, longer regimens, the implementation of the 6-month BPaLM regimen could improve the cost-effectiveness of care for individuals diagnosed with RR-TB, particularly in settings with a high burden of drug-resistant TB. Further research may be warranted to explore the impact and cost-effectiveness of shorter RR-TB regimens across settings with varied drug-resistant TB burdens and national income levels.

## Author summary

### Why was this study done?

- Drug resistance poses a major barrier to the effective treatment of tuberculosis, especially in Moldova and other post-Soviet states which have the highest levels of resistance in the world.

- Individuals with tuberculosis resistant to the key drug rifampicin face a worse prognosis, a longer and more expensive course of treatment, and more side effects than individuals with rifampicin-susceptible tuberculosis.

- Until recently, the standard of care for rifampicin-resistant tuberculosis (RR-TB) involved many drugs in combination, often given for 18 months or longer.

- The newer, 6-month "BPaLM" regimen is comprised of 4 drugs (bedaquiline, pretomanid, linezolid, and moxifloxacin) to which resistance levels are currently low, and while it was shown to be just as effective as the standard of care when health outcomes were measured at 72 weeks from treatment initiation, its effect on lifetime health outcomes, costs, and the acquisition of drug resistance was less clear.

### What did the researchers do and find?

- Using a mathematical model, we projected the lifetime health benefits and costs of the 6-month BPaLM regimen as compared to 9–18 month, standard of care treatments for RR-TB, and found that 6 months of BPaLM would be likely to provide similar health benefits, at lower cost.

- Compared to the standard of care, we also found that the 6-month BPaLM regimen could shorten the average duration of tuberculosis resistant to the drugs amikacin, bedaquiline, clofazimine, cycloserine, moxifloxacin, and pyrazinamide, while it may increase the average duration of tuberculosis resistant to delamanid and pretomanid.

- For individuals receiving BPaLM who had to stop taking the drug moxifloxacin, we found that it would likely be beneficial on both health and cost grounds to replace it with clofazimine, thereby topping the regimen back up to 4 drugs.

### What do these findings mean?

- Using conventional benchmarks for value-for-money, we estimated that 6 months of BPaLM would be a cost-effective approach for the treatment of RR-TB in Moldova, and potentially other post-Soviet countries.

- Though the impact of the 6-month BPaLM regimen on the spread of drug resistance is uncertain and not addressed directly by this study, this combination of newer drugs appears to achieve cure more quickly, thereby reducing the amount of time an individual is potentially infectious. This may be beneficial in fighting resistance to several drugs, even while it may increase the spread of resistance to others.

- Further studies may be warranted to explore how well these findings would translate to different global regions where health system capabilities, costs, and existing resistance patterns may differ.

## Introduction

Treatment for rifampicin-resistant tuberculosis (RR-TB) is complex, involving combinations of several drugs—many of which have substantial potential for toxicity—over a prolonged course of therapy. The 2022 WHO Guidelines for the treatment of drug-resistant tuberculosis recommended a shorter, 6-month regimen composed of bedaquiline, pretomanid, linezolid, and moxifloxacin (BPaLM) to treat RR-TB [1]. These guidelines updated earlier 2020 WHO Guidelines which recommended several treatment regimens, each comprising 4 to 7 drugs for 9 to 18 months or longer [2].

The evidence base for shorter regimens for RR-TB has been broadly positive, including results from observational studies [3,4], single-arm clinical trials [5,6], mathematical modeling analyses [7], and the recent multicenter open-label randomized controlled trial TB-PRACTE-CAL [8]. Although trial recruitment was stopped early on the recommendation of a planned, interim review by the study monitoring committee, the analysis suggested that 6 months of BPaLM was non-inferior to the standard of care with respect to treatment outcome (a

composite of death, treatment failure, treatment discontinuation, loss to follow-up, or recurrence) and was beneficial with respect to safety [8]. The adoption of shorter, simplified regimens may be further bolstered by the forthcoming publication of the results of the endTB trial [9–13], but in 2022 the absence of larger, confirmatory trials led to a conditional recommendation by the WHO. The pursuit of effective shorter treatment regimens is also driven by the desire to alleviate the considerable psychological and emotional toll of prolonged treatment for RR-TB. On top of drug side effects [14], many patients undergoing treatment for RR-TB experience stigma, depression, loss of self-esteem, and economic hardship from an inability to work [15]. Patients may lack access to sufficient psychological and financial supports [16–18], and this may be particularly hard for individuals with housing or employment instability, or substance use disorder [19].

The 2020 WHO Guidelines represent the existing standard of care in many settings. In addition to higher prices and supply constraints for newer drugs [20,21], it is expected that the rollout of the BPaLM regimen as part of the newer 2022 Guidelines may be delayed by concerns about comparative effectiveness and cost-effectiveness [20–24]. Implementation may also be met with concern over the emergence of drug resistance, particularly in settings with limited capacity to detect resistance to newer agents such as bedaquiline, pretomanid, and linezolid [25]; such capacity constraints are multifactorial, from the expense of investing in new technologies and associated laboratory workforce development, to supply chain interruptions and divergent political priorities [26,27]. The decision to implement the new 6-month BPaLM regimen will depend on setting-specific tradeoffs between regimen effectiveness, cost, the complexity of treatment decisions, and existing levels of resistance to anti-TB drugs in the population. Decision analysis provides a framework to analyze these tradeoffs, and a recent cost-effectiveness study using evidence from TB-PRACTECAL found that 6 months of BPaLM may reduce costs and improve health relative to the standard of care in several countries [28]. Our analysis builds on this work by focusing on longer term outcomes that are difficult to measure in a trial setting and by examining a wider range of testing and treatment approaches, including whether patients who must stop moxifloxacin (Mfx)—due to side effects or acquired resistance—should continue on BPaL alone or BPaL plus clofazimine (BPaLC) [25,28].

In this study, we investigated the health impact and cost-effectiveness a 6-month BPaLM regimen for the treatment of adults with pulmonary RR-TB, as compared to the standard of care. We considered a range of treatment strategies incorporating these 2 approaches, varying the timing and frequency of drug susceptibility testing (DST) as well as how regimens would be modified for individuals developing fluoroquinolone resistance (FQ-R). To estimate outcomes, we used a Markov microsimulation model parameterized with detailed genomic sequencing data describing specific patterns of initial drug resistance, and calculated the potential impact of each treatment strategy on length and quality of life as well as costs, accounting for the comparative effectiveness of the regimen used, risks of severe adverse events (SAEs) due to drug toxicity, and acquisition of resistance.

We conducted the analysis for the setting of Moldova, an upper-middle income post-Soviet country where the incidence rate of RR-TB is among the highest in the world, and where an estimated 33% of individuals newly diagnosed with TB have RR-TB, 10 times higher than the same proportion globally [29,30]. The reasons behind this are not fully understood, but it is thought that economic shocks following the breakup of the Soviet Union contributed to this picture in the region, along with early treatment discontinuation [31] and mass incarceration [32]. In current practice in Moldova, a multidisciplinary committee reviews the treatment course of every patient receiving treatment for RR-TB, and WHO treatment guidelines are closely adhered to [VC, DC]. Moldova also has developed a robust TB laboratory infrastructure, which provided a platform for recent genomic sequencing of culture-positive isolates

[33]. By harnessing this genomic resistance data, we hope to inform the cost-effectiveness of treatment in a country with a very high burden of RR-TB. We also explored the generalizability of our findings to settings with a different prevalence of initial FQ-R among RR-TB.

## Methods

### Strategies

We compared 8 treatment strategies, each reflecting a different approach to drug regimen choice and timing of DST (Table 1). In 2 strategies, drug regimens aligned with the standard of care as defined by the 2020 WHO Guidelines [2], with all individuals starting on a WHO

**Table 1. Key features of the modeled RR-TB treatment strategies.**

| Strategy No. | Guidelines informing the strategy | Drug regimen | Regimen duration | For BPaLM-based strategies only, prescribed regimen for those who discontinue Moxifloxacin | Replacement drugs, in order, for all other discontinuations | DST for second-line drugs (MGIT) at treatment initiation | Routine frequency of subsequent DST for second-line drugs | Indications for drug discontinuation | Length of regimen extension, if necessary* |
|---|---|---|---|---|---|---|---|---|---|
| 1 | 2022 WHO Guidelines | BPaLM | 6 months | BPaLC | Clofazimine, Cycloserine | Yes | 4 months | Immediately following: • Resistance identified on DST • Grade 4–5 Severe Adverse Event | 6 months |
| 2 | 2022 WHO Guidelines | BPaLM | 6 months | BPaLC | | Yes | 1 month | | |
| 3 | 2022 WHO Guidelines | BPaLM | 6 months | BPaL | | Yes | 4 months | | |
| 4 | 2022 WHO Guidelines | BPaLM | 6 months | BPaL | | Yes | 1 month | | |
| 5 | 2022 WHO Guidelines | BPaLM | 6 months | BPaLC | | No | 4 months | | |
| 6 | 2022 WHO Guidelines | BPaLM | 6 months | BPaL | | No | 4 months | | |
| 7 | 2020 WHO Guidelines (standard of care) | Start treatment with WHO longer regimen (bedaquiline, clofazimine, linezolid, moxifloxacin), await second-line DST. **If FQ-R,** continue on WHO Longer regimen, i.e.: | | | | Yes | 4 months | | |
| | | Bedaquiline, Clofazimine, Linezolid, Cycloserine | 18 months | n/a | Ethambutol, Delamanid, Pyrazinamide, Amikacin, Ethionamide | | | | |
| | | **If FQ-S,** switch to 2020 WHO shorter, all-oral bedaquiline containing regimen: | | | | | | | |
| | | Bedaquiline, Clofazimine, Ethambutol, Ethionamide, Isoniazid, Moxifloxacin, Pyrazinamide | 9 months | n/a | Delamanid, Cycloserine | | | | |
| 8 | 2020 WHO Guidelines (standard of care) | As for Strategy No. 7 | | | | Yes | 1 month | | |

*Regimen extensions were implemented for those who had not yet successfully completed treatment. While the 2020 WHO Guidelines recommended the BPaL regimen in specific situations, none of the modeled cohort met the inclusion criteria to receive BPaL under those strategies.

BPaL, bedaquiline, pretomanid, linezolid; BPaLC, bedaquiline, pretomanid, linezolid, clofazimine; BPaLM, bedaquiline, pretomanid, linezolid, moxifloxacin; DST, drug susceptibility test; FQ-R, fluoroquinolone resistant; FQ-S, fluoroquinolone susceptible; MGIT, mycobacterial growth indicator tube; WHO, World Health Organization.

longer regimen while awaiting the results of second-line DST by mycobacterial growth indicator tube (MGIT) to fluoroquinolones and injectables. FQ-R identified via MGIT was assumed to result in the continuation of an 18-month WHO longer regimen, with refinements as necessary based on DST. If fluoroquinolone susceptibility (FQ-S) was detected, treatment was switched to a 9-month regimen (S1 Fig). Under 1 standard of care strategy (Strategy (7)), we set the frequency of second-line DST to every 4 months as per the minimum guideline-recommended interval [1], and in another (Strategy (8)), we increased this to a monthly frequency. While the 2020 WHO Guidelines did not prescribe exactly one combination of drugs for each scenario, we adopted a single combination of drugs for each situation for tractability, based on our best interpretation of the guideline's hierarchy of group A, B, and C drugs (S1 Fig).

The remaining 6 strategies were modeled on the 2022 WHO Guidelines [34] with 6-month BPaLM-based regimens. In 3 of these strategies, individuals having to stop moxifloxacin (because of a SAE or because resistance was detected on DST) were continued on BPaL alone, as recommended by the 2022 Guidelines. In the remaining 3, they continued on BPaLC. The remaining differences between these 6 strategies depended on the prescribed schedule of DST to second-line drugs; in 2 of these strategies, we explored the potential impact of omitting routine second-line DST at treatment initiation (Table 1).

## Population and data

We modeled a cohort of individuals aged 15 years and older diagnosed with RR-TB in Moldova. For each individual, their age and the resistance profile of the strain of *Mycobacterium tuberculosis* causing infection were informed by publicly available genomic sequencing data from Moldova [35]. These data comprised single-strain *M. tuberculosis* samples collected in 2018 and 2019; a full description has been provided by Yang and colleagues [30]. We assumed that a mutation associated with resistance conferred full resistance to that drug. Conversely, *M. tuberculosis* strains lacking relevant resistance mutations were assumed to be fully susceptible to the respective drugs. We excluded data for rifampicin-susceptible strains (S2 Fig) leaving 674 distinct samples from which we simulated the modeled population. The proportion of isolates with resistance to each drug is shown in S5 Fig. This analysis used publicly available data only and did not require ethical approval.

## Model

We used a Markov microsimulation model to simulate lifetime outcomes for a cohort of 10,000 individuals. Individuals in the model were simulated by random draws from the genomic sequencing dataset, with replacement. They were each assigned a drug regimen based on the modeled strategy (Table 1). Individuals then were assumed to transition between 4 health states: (1) receiving TB treatment; (2) TB disease—not receiving treatment; (3) cured post-treatment; and (4) dead (S4 Fig). Within each Markov state, individual events were tracked including true cure as a result of treatment or self-cure, the occurrence of grade 4–5 SAEs, second-line DST, changes to the drug regimen, loss to follow-up, relapse, death, and the evolution of drug resistance within that individual's strain of *M. tuberculosis* to each of 12 anti-TB drugs. Extensions to the treatment regimen were implemented for those not observed to have successfully completed treatment.

For each individual in any given month, a health-related quality of life weight was assigned based on that individual's month of treatment and any grade 4–5 SAEs experienced. In this way, we sought to capture how drug therapy may impact quality of life positively (treatment gradually eases TB-related symptoms) as well as negatively (treatment may be associated with toxicity) [36]. While the range of SAEs resulting from TB treatment are of many varying

**Table 2. Key model parameters.**

| Parameter | Point estimate | Distribution | Source(s) | Notes |
|---|---|---|---|---|
| Rate of death from untreated TB, annual | 0.389 | Published point estimate (median) and 95% CrI (0.335–0.449) modeled as Lognormal (mu −0.9442, sigma 0.0763) | Ragonnet R, et al. 2020 [37] | Applied to those with TB no longer receiving treatment (i.e., those LTFU and those who appeared to successfully complete treatment but had not been truly cured). |
| Mortality rate ratio for those who are cured, compared to background mortality | 3.07 | Published point estimate and 95% CI (2.12, 4.45) modeled as Lognormal (mu 1.122, sigma 0.1889) | Romanowski K., et al. 2019 [38] | Estimate for pulmonary TB. |
| Rate of self-cure, annual | 0.231 | Published point estimate and 95% CrI (0.177, 0.288) modeled as Lognormal (mu −1.465, sigma 0.136) | Ragonnet R, et al. 2020 [37] | Applied to those with TB no longer receiving treatment (i.e., those LTFU and those who appeared to successfully complete treatment but had not been truly cured) and the first 2 months of treatment. |
| Probability of all-cause death for WHO longer regimen, MDR only (excluding XDR), at 21 months | 0.080 | N/A | Bastos M. L., et al. 2017 [39] | To convert to a monthly estimate for disease-specific mortality, we assumed a 21 month regimen duration. Further detail in S2 Appendix. |
| Mortality rate among those who are not cured but on treatment, MDR-TB only (excluding XDR-TB), monthly | 0.00536 | Beta (mean 0.00536, s.d. 0.00178)* | Bastos M. L., et al. 2017 [39] | See S2 Appendix. |
| Probability of observed success for a fully effective WHO longer regimen, MDR-TB only (excluding XDR), at 21 months | 0.640 | Published point estimate and 95% CI (0.63, 0.65) modeled as Beta (mean 0.64, s.d. 0.0051) | Bastos M. L., et al. 2017 [39] | To convert to a monthly cure rate for standard of care strategies, we assumed a 21-month regimen duration. We used this parameter to inform the comparative effectiveness of a fully effective regimen of 4 drugs (i.e., a regimen composed of 4 drugs to which the individual's strain of *M.tb.* is truly susceptible). See also S2 Appendix. |
| Hazard rate ratio of cure for each effective drug in the regimen (relative to one fewer effective drugs) | 1.65 | Published point estimate and 95% CI (1.48, 1.84) modeled as Lognormal (mu 0.501, sigma 0.056) | Yuen, CM. et al. 2015 [40] | Applied to a maximum of 4 drugs (i.e., there was no further increase in the monthly cure rate for 5 drugs compared to 4). |
| Hazard rate ratio of cure for the BPaLM regimen as compared to the SOC | 1.59 | Published point estimate and 95% CI (1.18, 2.14) modeled as Lognormal (mu 0.453, sigma 0.147) | Nyang'wa, B.-T. et al. 2022 [8] | The referenced estimate is based on the outcome of time to sputum culture conversion. We assumed the same relationship holds for the rate of true cure and explored this assumption in sensitivity analysis. |
| Probability of acquiring resistance to a drug, conditional on treatment with a regimen of 4 or more effective drugs, over 6 months | 0.008 | Published point estimate and 95% CI (0.005, 0.010) modeled as Beta (mean 0.008, s.d. 0.0015) | Lew W. et al. 2008 [41] | We defined an "effective" drug as one to which the strain of *M. tuberculosis* was susceptible. We used the published estimate to produce a monthly rate of resistance acquisition, which was constant conditional on the number of effective drugs. |
| Health-related quality of life weight during treatment | 0.750–0.990 | See S1 Table | Bauer M., et al. 2015 [36] | The health-related quality of life weight varied within this range, depending on the month of treatment. Please see S1 Table for more details. |
| Health-related quality of life weight decrement for grade 4–5 severe adverse event | 0.056 | Published point estimate and S.E.M (0.006) modeled as Beta (mean 0.056, s.d. 0.006) | Takahara M, et al. 2019 [42] | This decrement was deducted for each grade 4–5 SAE experienced and was assumed to be lifelong. For parsimony, we assumed this mild but lifelong decrement reflected the average consequence among the many different possible grade 4–5 SAEs that could be experienced. |
| Monthly cost of drug regimens in BPaLM-based strategies (strategies (1)–(6)) (2022 USD) | 130.36–229.83 | See S1 Table | Stop TB Partnership Global Drug Facility Medicines Catalog [20] | Please see S1 Table for the cost input parameter for each individual drug, as well as all other cost parameters. |
| Monthly cost of drug regimens in SOC-based strategies (strategies (7) and (8)) (2022 USD) | 88.91–197.55 | See S1 Table | Stop TB Partnership Global Drug Facility Medicines Catalog [20] | |

*For this parameter, there was no readily available measure of dispersion; we assumed a standard deviation equal to one third of the mean.

Please see S1 Table for the full set of model parameter descriptions, and S2 Table for the probabilities of loss to follow up by month of treatment.

BPaLM, bedaquiline, pretomanid, linezolid, moxifloxacin; CI, confidence interval; CrI, credibility interval; LTFU, lost to follow up; M. tb.; *Mycobacterium tuberculosis*; MDR, multidrug-resistant; NHB, net health benefit; QALY, quality-adjusted life year; RR-TB, rifampicin-resistant tuberculosis; TB tuberculosis; UI, uncertainty interval; USD, United States dollars; WHO, World Health Organization; WTP, willingness-to-pay; XDR, extensively drug resistant.

durations and degrees of impact on quality of life, we accounted for these events in a simplified way by modeling the risk of a typical grade 4–5 SAE during the first 3 months of exposure to each drug, with each grade 4–5 SAE conferring a small but lifelong deduction in quality of life (Tables 2 and S1). Grade 4–5 SAEs and diagnosed resistance constituted lifetime contraindications to the relevant drug, and replacements were made according to the modeled strategy (Table 1).

Each month, we tracked the drug regimen and the true resistance profile of each individual's strain of *M. tuberculosis*. The number of effective drugs in a regimen was defined as the sum of all drugs being received, minus those drugs to which the strain of *M. tuberculosis* was resistant. The estimate for the hazard rate ratio (HRR) of cure in BPaLM-based strategies as compared to the standard of care was modeled as the estimate for sputum culture conversion from TB-PRACTECAL, conditional on the number of effective drugs in the regimen, up to a maximum of 4 (i.e., 4 effective drugs conferred a higher monthly cure rate than 3, but 5 or more effective drugs did not confer a higher monthly cure rate than 4) [8]. We varied this parameter in sensitivity analysis. S3 Fig displays the modeled rate of acquisition of new resistance to each drug, which was also conditioned on the number of effective drugs, to a maximum of 4. S1 Table details the derivation and values for these and all other model parameters. DST was performed at a frequency informed by the strategy (Table 1), with sensitivity and specificity incorporated for each (S1 Table). Additional detail on model structure is provided in S1 Appendix.

## Outcomes

The primary health outcome was measured in quality-adjusted life years (QALYs), a conventional approach in cost-effectiveness analysis [43,44]. This approach integrates the impacts of the treatment strategies on both length and quality of life. For each modeled individual in each month, we assigned health-related quality of life weights on a scale from 0 (dead) to 1 (perfect health) as described above, and multiplied the weight by 1/12 to obtain the QALYs accrued for that month. The total QALYs were calculated by summing all the month-specific QALYs accrued over each modeled individual's lifetime.

We measured the impact on drug resistance by summing for each individual, and for each of 13 anti-TB drugs, the number of months they experienced TB disease with resistance to that drug. We then calculated 3 summary measures for the impact on drug resistance. In the first, we calculated the mean duration with resistance to each drug for the entire cohort by aggregating the time with resistance across the whole cohort for each drug, then dividing by the size of the starting cohort. Second, we calculated the mean duration of untreated TB disease with resistance to each drug by summing the time with resistance only among those individuals in Markov state (2)—TB disease no longer receiving treatment—and again averaging across the starting cohort. These measures were designed to reflect the potential relevance of the policies for the transmission of drug resistance. We calculated both because—for individuals no longer receiving treatment—there could be a higher risk that *M. tuberculosis* would transmit to another host, compared to the cohort as a whole. Third, we calculated the lifetime cumulative incidence of acquiring resistance to each drug, per individual in the cohort.

As a set of secondary health outcomes, we calculated the number of grade 4–5 SAEs experienced per patient to each of the drugs, and total life years (LYs, i.e., not weighted by health-related quality of life). To permit the validation of our model results, we also tracked 2 types of shorter-term outcomes at 6 months, 12 months, and 17 months (i.e., 72 weeks, the trial endpoint in TB-PRACTECAL): (1) the proportion of individuals who had experienced the end-of-treatment (EOT) outcomes of success, failed by treatment, lost to follow-up (LTFU), and death, as would typically be reported programmatically to the WHO (S6 Fig); and (2) a

composite unfavorable outcome, including death, LTFU, failed by treatment, and grade 4–5 SAEs, based on the primary outcome in TB-PRACTECAL [8].

We measured the total costs under each strategy from a societal perspective in 2022 United States dollars ($) as the sum of direct medical, direct non-medical, and indirect costs accruing in each period. Direct medical costs (i.e., those arising directly from the consumption of healthcare goods and services) were calculated by adding the costs of the drugs received, laboratory culture and DST to second-line drugs, a baseline healthcare resource utilization in the form of inpatient and outpatient services, and the cost of LTFU tracing. Direct non-medical and indirect costs were informed by published estimates for Moldova [45]. Each grade 4–5 SAE was accompanied by a utilization cost for inpatient and outpatient services (S1 Table). Direct non-medical costs (e.g., transportation) and indirect costs (e.g., productivity losses) accrued for every additional month on treatment. The indirect costs also accrued for those LTFU prior to cure. Productivity losses secondary to early mortality were not included in total costs and were aggregated separately.

Undiscounted values were calculated for all outcomes. For QALYs and total costs only, discounted values were also calculated using an annual discount rate of 3%.

**Cost-effectiveness analysis.** First, we ruled out dominated strategies (i.e., those strategies that were both more expensive and provided fewer QALYs on average than a linear combination of other strategies). We then calculated the relevant incremental cost-effectiveness ratios (ICERs; a measure of the additional cost required to produce one additional QALY, as compared to the next cheapest, non-dominated strategy). We identified the cost-effective strategy as that with the greatest health gains, subject to the constraint that—in order to provide value for money—the ICER must be below the willingness-to-pay (WTP) threshold [43,46]. Lower ($4,700 per QALY) and higher ($7,021 per QALY) benchmarks for these thresholds in Moldova were based on published estimates using an opportunity cost approach [47], updated to 2022 USD (S1 Table). As the interpretation of ICERs may be challenging in some circumstances [48], we also calculated the net health benefit (NHB) of each strategy (see S1 Appendix), with the cost-effective strategy identified as that with the highest NHB [43]. This is mathematically equivalent to the ICER approach. The CHEERS checklist is included in S1 Checklist [49].

**Budget impact.** In order to account for the potential consequences of implementing 6 months of BPaLM on the national TB program budget in Moldova, we tracked the subset of aforementioned cost outcomes borne by the TB program. We organized these costs under the following categories: drugs, laboratory tests, routine inpatient and outpatient care, and non-routine inpatient and outpatient care (i.e., care stemming from the treatment of grade 4–5 SAEs, for adjustment of a regimen following the detection of resistance on DST or for LTFU tracing). The estimated budget impact was calculated for each year over a 5-year period, scaled to the annual number of case notifications of RR-TB in Moldova.

## Statistical analysis

We estimated results via individual-level microsimulation, with lifetime outcomes for each of 10,000 individuals simulated for each of the diagnostic and treatment strategies described above.

**Sensitivity analyses.** Probabilistic sensitivity analysis (PSA) was conducted to account for uncertainty by constructing distributions for model input parameters (S1 Table). In a second-order Monte Carlo simulation, we drew 1,000 parameters sets from the distributions. For each parameter set, the 10,000 individuals were simulated through each strategy, and a set of results was calculated. Finally, point estimates for each outcome were calculated as the mean of these 1,000 second-order simulations, and 95% uncertainty intervals (UIs) were constructed using the 2.5th and 97.5th centiles [50]. Point estimates and 95% UIs were also calculated for the

differences between leading 6-month BPaLM-based and SOC-based strategies, and *p*-values were constructed from the empirical cumulative distribution function of those differences. Further detail is provided in S1 Appendix.

Some important model parameters have substantial uncertainty. We performed one-way sensitivity analyses on 2 of these key inputs to understand the relationship with study outcomes. First, we varied the main comparative effectiveness estimate for cure across the uniform distribution (1.00, 2.14). Next, we varied the prevalence of FQ-R among individuals with diagnosed RR-TB across the uniform distribution (0%, 40%) to aid the generalization of results to settings with a different prevalence of FQ-R.

**Validation.**   We validated the modeled EOT outcomes to estimates reported to WHO over the period 2010 to 2019. We also validated the composite of unfavorable outcome at 72 weeks against the findings of TB-PRACTECAL [8]. Further detail is provided in the S1 Appendix.

**Software.**   The simulation was conducted in TreeAge Pro Healthcare 2023 [51] and figures were made in R [52] using several packages [53–61]. TreeAge and R code files are available in a repository [62].

## Results

### Health effects, costs, and cost-effectiveness

Health effects, costs, and cost-effectiveness results for all strategies are presented in Table 3 and Fig 1. Among the 6-month BPaLM strategies, the highest health benefits were estimated under Strategy (1) (BPaLC if Mfx stopped, second-line DST upfront, then repeated at 4 monthly intervals), with undiscounted QALYs of 14.75 (95% UI: [12.76, 16.54]). The 2 standard of care strategies (Strategies (7) and (8)) both were estimated to produce slightly more QALYs than Strategy (1), with less than 0.01 undiscounted QALYs between them on average. The Life Years (unadjusted for health-related quality of life) estimated under each strategy are displayed in S4 Table.

Strategy (5) (6-months BPaLM, second-line DST at 4 months and then every 4 months, BPaLC if Mfx stopped) had the lowest undiscounted lifetime total costs ($8412, 95% UI: [6469, 10991]), followed by Strategy (1) and Strategy (2) (Table 3).

Compared to 6-month BPaLM-based strategies where BPaLC was used if Mfx had to be stopped, strategies continuing only the three-drug regimen BPaL (Strategies (3), (4), and (6)) were estimated to result in worse overall health and additional lifetime total costs. The frequency of second-line DST did not lead to large differences in health or cost outcomes (Fig 1).

We compared ICERs to current cost-effectiveness criteria for Moldova, with the willingness-to-pay for health improvements assumed to fall between $4,700 and $7,021 per QALY gained. According to this approach, Strategy (1) (6-months BPaLM, DST upfront then every 4 months, BPaLC if Mfx stopped) was the most cost-effective strategy, with an ICER of $4,375 per QALY.

Strategy (7) was potentially cost-effective, but only with a willingness to pay over $56,100 per additional QALY, far higher than the upper bound threshold. In Fig 1B, we show the probability that each strategy is the most cost-effective for given cost-effectiveness thresholds.

Strategies (1), (2), and (5) had the highest probabilities of being cost-effective. All 3 were 6-month BPaLM strategies where BPaLC was continued for those stopping Mfx, and differed only based on the schedule of routine DST. Taken together, the probability that one of these strategies would be most cost-effective was at least 93% across the range of cost-effectiveness thresholds for Moldova.

For simplicity, we henceforth make comparisons between the leading (i.e., most cost-effective based on point estimates) 6-month BPaLM-based and standard of care-based strategies:

**Table 3. Costs, health impacts, and cost-effectiveness of RR-TB treatment strategies.**

| | Strategy description | | | | Cost | | Health impact | | Cost-effectiveness | | | | |
|---|---|---|---|---|---|---|---|---|---|---|---|---|---|
| Strategy name | Alternative regimen if Mfx stopped (BPaLM-based strategies only) | 2nd line DST at treatment initiation | Routine frequency of subsequent 2nd line DST | Undiscounted total cost (2022 USD) | Discounted total cost (2022 USD) | Undiscounted QALYs | Discounted QALYs | Incremental discounted total cost (2022 USD) | Incremental discounted QALYs | ICER | NHB, lower bound WTP (QALYs) | NHB, upper bound WTP (QALYs) |
| (5) 6 months BPaLM | BPaLC | No | Every 4 months | 8,412 (6,469, 10,974) | 8,153 (6,279, 10,592) | 14,745 (12.72, 16.55) | 10,494 (9.26, 11.5) | (comparator) | (comparator) | (comparator) | 8.759 (7.42, 9.87) | 9.333 (8.07, 10.39) |
| **(1) 6 months BPaLM** | **BPaLC** | **Yes** | **Every 4 months** | **8,424 (6,469, 10,991)** | **8,167 (6,299, 10,629)** | **14,750 (12.76, 16.54)** | **10,497 (9.28, 11.52)** | **14 (−170, 205) $p = 0.876$** | **0.0032 (−0.16, 0.19) $p = 0.996$** | **4,375** | **8.759 (7.46, 9.90)** | **9.334 (8.09, 10.43)** |
| (2) 6 months BPaLM | BPaLC | Yes | Every 1 month | 8,663 (6,713, 11,219) | 8,398 (6,518, 10,767) | 14,753 (12.74, 16.49) | 10,500 (9.28, 11.50) | -- | -- | Dominated* | 8.713 (7.40, 9.83) | 9.304 (8.05, 10.38) |
| (6) 6 months BPaLM | BPaL only | No | Every 4 months | 9,059 (6,926, 11,876) | 8,723 (6,707, 11,394) | 14,405 (12.44, 16.10) | 10,275 (9.07, 11.24) | -- | -- | Dominated | 9.033 (7.78, 10.07) → 8.419 (7.10, 9.51) | |
| (3) 6 months BPaLM | BPaL only | Yes | Every 4 months | 9,085 (6,935, 11,925) | 8,750 (6,703, 11,436) | 14,414 (12.42, 16.17) | 10,280 (9.10, 11.29) | -- | -- | Dominated | 8.419 (7.07, 9.55) | 9.034 (7.80, 10.09) |
| (4) 6 months BPaLM | BPaL only | Yes | Every 1 month | 9,391 (7,201, 12,126) | 9,039 (6,950, 11,635) | 14,408 (12.42, 16.16) | 10,276 (9.08, 11.28) | -- | -- | Dominated | 8.353 (7.04, 9.50) | 8.989 (7.71, 10.08) |
| (7) Standard of care | N/A | Yes | Every 4 months | 11,936 (9,403, 15,102) | 11,534 (9,088, 14,619) | 14,832 (13.00, 16.54) | 10,557 (9.46, 11.54) | 3,366 (1,465, 5,742) $p = {<}0.001$ | 0.0600 (−0.32, 0.49) $p = 0.79$ | 56,100 | 8.103 (6.88, 9.19) | 8.915 (7.77, 9.92) |
| (8) Standard of care | N/A | Yes | Every 1 month | 12,232 (9,727, 15,435) | 11,816 (9,395, 14,912) | 14,836 (12.98, 16.56) | 10,560 (9.46, 11.55) | 282 (24, 550) $p = 0.028$ | 0.0025 (−0.16, 0.18) $p = 0.996$ | 112,800 | 8.045 (6.83, 9.10) | 8.877 (7.73, 9.87) |

Strategies listed in order of increasing discounted total cost. Lower bound willingness-to-pay = 4,700 USD/QALY; upper bound willingness-to-pay = 7,021 USD/QALY. Dominated strategies are those that were both more costly and resulted in poorer health than at least one other strategy, based on point estimates. For non-dominated strategies (i.e., strategy numbers 1, 5, and 7), the cheapest (Strategy 5) is listed as the comparator. For Strategies 1, 7, and 8, incremental discounted total cost and incremental discounted QALYs were calculated relative to the next cheapest, non-dominated strategy. The most cost-effective strategy is highlighted in **bold** text. Mean values are shown with accompanying 95% UIs in parentheses.

*Strategy (2) was dominated by extended dominance; ICER = 77,000 USD/QALY.

BPaL, bedaquiline, pretomanid, linezolid; BPaLC, bedaquiline, pretomanid, linezolid, clofazimine; BPaLM, bedaquiline, pretomanid, linezolid, moxifloxacin; DST, drug susceptibility testing; ICER, incremental cost-effectiveness ratio; Mfx, moxifloxacin; NHB, net health benefit; QALY, quality-adjusted life year; RR-TB, rifampicin-resistant tuberculosis; UI, uncertainty interval; USD, United States dollars; WTP, willingness-to-pay.

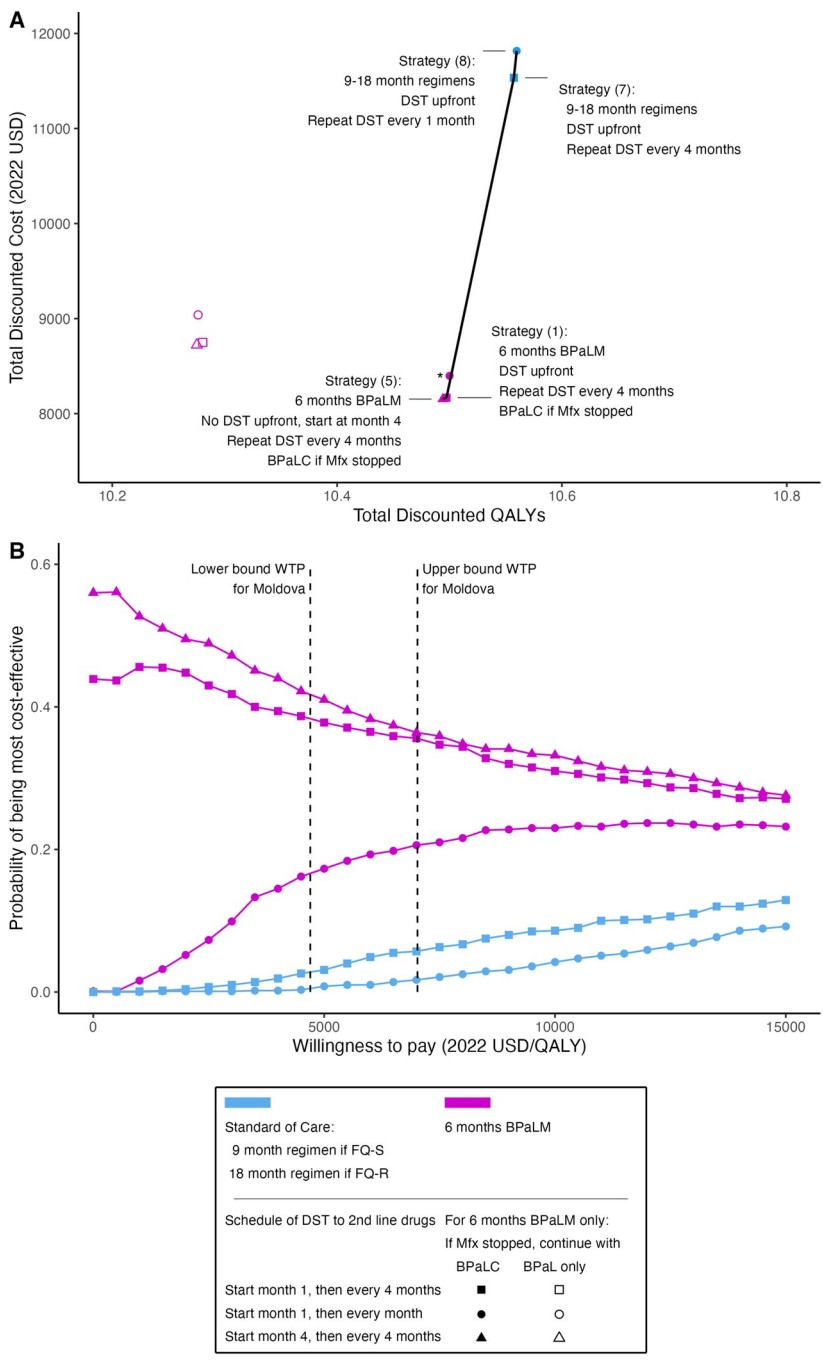

**Fig 1. Six months of BPaLM is a cost-effective treatment approach for RR-TB.** (A) The cost-effectiveness plane shows point estimates for the discounted total costs and discounted QALYs under each modeled strategy. These are calculated as the mean of all simulation runs (1,000 second-order Monte Carlo simulations, each with 10,000 individual patient simulations). Non-dominated strategies are labeled, and the efficient frontier (black lines) connects the non-dominated strategies based on point estimates. (B) The cost-effectiveness acceptability curve displays the probability that each modeled strategy is the most cost-effective strategy at different levels of WTP. This probability is calculated as the proportion of 1,000 second-order Monte Carlo simulations where the respective strategy was optimal, given the value for WTP. Strategies were excluded from the cost-effectiveness acceptability curve if they were not cost-effective in any of the simulations (these were Strategies 3, 4, and 6, where BPaL only was used if Mfx had to be stopped under a BPaLM regimen). Vertical dashed lines mark the lower and upper bounds of the WTP thresholds for Moldova. *Strategy (2) (6 months BPaLM, BPaLC if Mfx discontinued, DST upfront then every 1 month) was close to the efficient frontier but was dominated by extended dominance based on point estimates. BPaL, bedaquiline, pretomanid, linezolid; BPaLC, bedaquiline, pretomanid, linezolid, clofazimine; BPaLM, bedaquiline, pretomanid, linezolid,

moxifloxacin; DST, drug susceptibility testing; FQ-R, fluoroquinolone-resistant; FQ-S, fluoroquinolone-susceptible; Mfx, moxifloxacin; QALY, quality-adjusted life year; USD, United States dollars; WTP, willingness-to-pay.

Strategy (1) and Strategy (7), respectively. The category-specific costs for these strategies are shown in Fig 2, and the incremental cost-effectiveness for this one-to-one comparison in S7 Fig.

Compared to the standard of care (Strategy (7)), the incremental NHB of 6 months BPaLM (Strategy (1)) was 0.656 QALYs; (95% UI [−0.091, 1.383] $p$ = 0.082) at the lower bound WTP and 0.419 QALYs; (95% UI [−0.206, 0.994] $p$ = 0.166) at the upper bound WTP.

## Drug resistance

When counting time with resistance across the entire cohort, compared to Strategy (7), Strategy (1) was associated with a nonsignificant change in the mean duration of RR-TB of −1.10 months; (95% UI [−4.07, 2.28] $p$ = 0.486) (Fig 3 and S3 Table). Strategy (1) was estimated to increase the mean duration with resistance to pretomanid by 0.55 months; (95% UI [0.20, 1.05] $p$ < 0.001) and delamanid by 0.54 months; (95% UI [0.18, 1.04] $p$ = 0.002) (Fig 3 and S3 Table).

In contrast, Strategy (1) decreased the duration with resistance for several drugs: The mean change was −2.21 months for moxifloxacin (95% UI [−3.39, −1.02] $p$ < 0.001), −2.28 months for pyrazinamide (95% UI [−4.02, −0.52] $p$ = 0.016), −1.31 months for clofazimine (95% UI [−1.94, −0.80] $p$ < 0.001), −0.92 months for bedaquiline (95% UI [−1.48, −0.49] $p$ < 0.001), −0.95 months for cycloserine (95% UI [−1.38, −0.62] $p$ < 0.001), and −0.40 months for amikacin (95% UI [−0.79, −0.06] $p$ = 0.022) (Fig 3 and S3 Table). When measuring time with resistance only among those with active, untreated RR-TB, or when measuring lifetime cumulative incidence of resistance, the findings revealed a similar picture (Fig 3 and S3 Table).

## Secondary outcomes

Under Strategy (7), the mean number of grade 4–5 SAEs ever experienced per individual was 0.265 (95% UI: 0.233, 0.300). Strategy (1) resulted in a mean number of grade 4–5 SAEs of 0.237 (95% UI [0.197, 0.284]), conferring a decrease of 0.028 grade 4–5 SAEs per person (95% UI [−0.012, 0.063] $p$ = 0.17) over the course of treatment. Fig 4 displays the proportion ever experiencing a grade 4–5 SAE to each drug; the point estimates were lower for Strategy (1) than for Strategy (7) for all drugs except linezolid and pretomanid.

When health benefits were measured using life years unadjusted for health-related quality of life, Strategy (7) again conferred a slightly higher life expectancy than Strategy (1) on expectation. Also consistent with the primary QALY-based outcomes, the lowest life expectancy was estimated for Strategies (3), (4), and (6) (BPaLM-based strategies where BPaL was continued in the event of Mfx being stopped) (S4 Table).

For the shorter-term endpoints of 6 months, 12 months, and 17 months (i.e., 72 weeks) from treatment initiation, we found that Strategy (1) resulted in a reduction in the composite unfavorable outcome compared to Strategy (7). The reduction was not significant when using the TB-PRACTECAL aligned definitions for unfavorable outcomes, but was significant and larger in magnitude when using WHO-based definitions (S5 Table and S9 Fig).

Compared to Strategy (7), Strategy (1) would be expected to save Moldova's national TB program budget $7.1 million (95% UI: [1.3 million, 15.4 million] $p$ = 0.002) over the 5-year period from implementation (S6 Table).

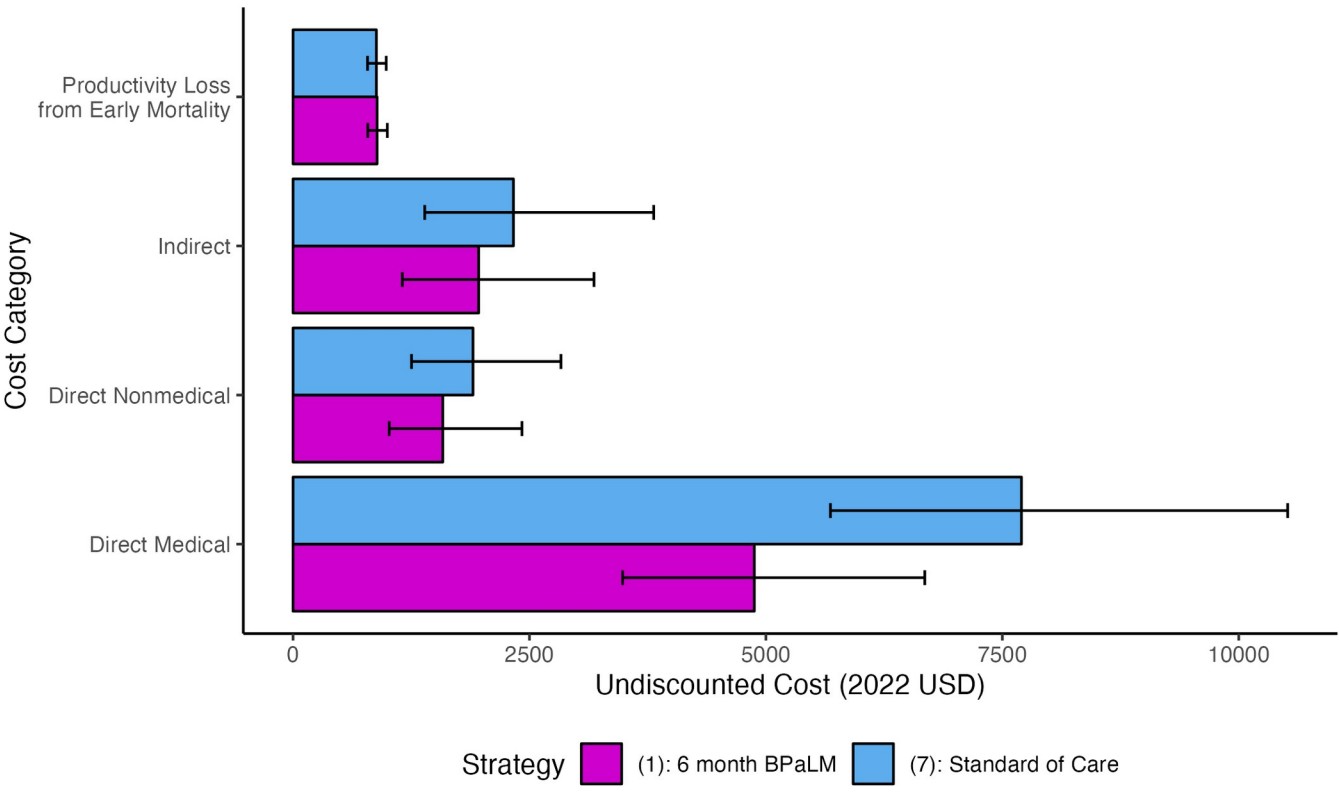

**Fig 2. Lifetime costs for 6-month BPaLM and standard of care strategies by category.** Undiscounted lifetime costs per individual for Strategy (1) (6 months BPaLM, DST upfront, repeat DST every 4 months, BPaLC if Mfx stopped) as compared to Strategy (7) (standard of care 9- to 18-month regimens based on results of upfront DST, repeat DST every 4 months). The bars show the mean model outcomes for each cost category, with error bars representing 95% UIs. BPaLC, bedaquiline, pretomanid, linezolid, clofazimine; BPaLM, bedaquiline, pretomanid, linezolid, moxifloxacin; DST, drug susceptibility testing; NHB, net health benefit; UI, uncertainty interval; USD, United States dollars; WTP, willingness-to-pay.

### Sensitivity analyses

Figs 5 and S8 show how results change for different values of the HRR of cure and the initial prevalence of FQ-R, for Strategy (1) as compared to Strategy (7). In these results, Strategy (1) was estimated to be cost-effective (i.e., had a positive NHB) compared to Strategy (7) across the range of values used for these parameters. Similarly, total costs were lower for Strategy (1) compared to Strategy (7) across the range of values assessed. Health outcomes (QALYs and LYs) were sensitive to the value of the HRR for cure for the BPaLM regimen as compared to standard of care regimens. For low values of the HRR (HRR = 1), Strategy (1) was estimated to lead to a mean 0.90 reduction in QALYs. For high values (HRR = 2), Strategy (1) would lead to a mean 0.35 gain in QALYs. All data files containing these results are available in a repository [62].

### Discussion

In this study, we assessed the potential health impact and cost effectiveness of a 6-month BPaLM regimen for treating RR-TB in a setting with a high prevalence of drug resistance. Compared to strategies using 9- to 18-month regimens based on the 2020 WHO treatment guidelines for drug-resistant TB, we found the 6-month BPaLM regimen would be cost-effective across a range of WTP thresholds, with substantial reductions in the duration and cost of treatment, but little expected change in health outcomes. Though there was considerable overlap between some of the 6-month BPaLM implementation scenarios, there was a clear lead for

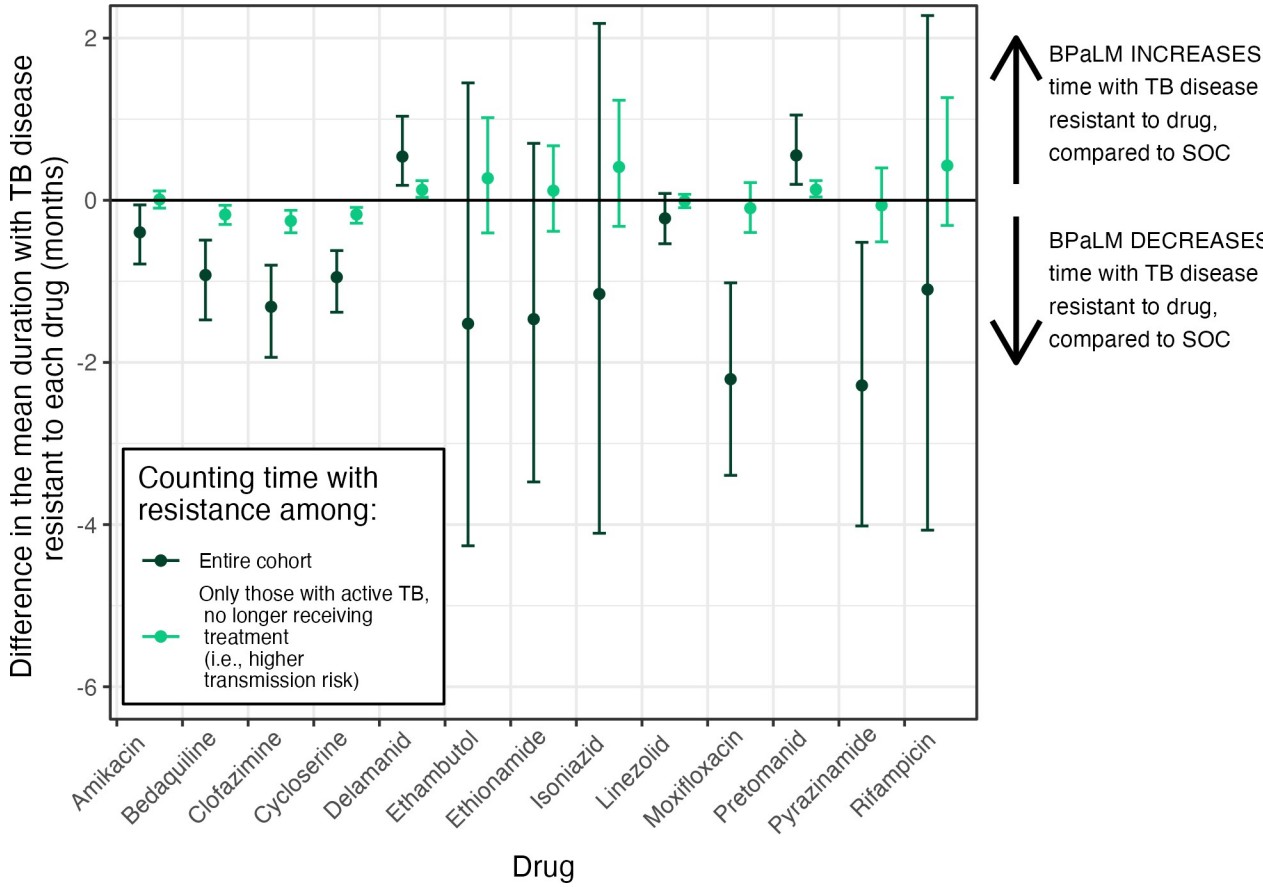

**Fig 3. Impact of 6 months BPaLM on duration of resistance to key anti-TB drugs.** Results are shown for Strategy (1) (6 months BPaLM, DST upfront, repeat DST every 4 months, BPaLC if Mfx stopped) as compared to Strategy (7) (standard of care 9- to 18-month regimens based on results of upfront DST, repeat DST every 4 months). For each drug, 2 estimates are provided: counting time with resistance at any point until the individual is truly cured (dark green), and counting time with resistance only while an individual has TB disease but is not being treated (light green). Both estimates are provided per individual, averaged over the same denominator of the entire cohort initiating treatment, and 95% UIs are shown by the accompanying error bars. BPaLC, bedaquiline, pretomanid, linezolid, clofazimine; BPaLM, bedaquiline, pretomanid, linezolid, moxifloxacin; DST, drug susceptibility testing; Mfx, moxifloxacin; SOC, standard of care; TB, tuberculosis; UI, uncertainty interval.

strategies where clofazimine was used to "top up" the regimen if moxifloxacin had to be discontinued because of a grade 4–5 SAE or resistant DST result, compared to continuing on the three-drug BPaL regimen alone. Holding the drug regimen constant, the frequency of second-line DST (to fluoroquinolones and injectables only, using MGIT) did not result in substantial differences to health or cost outcomes.

Our findings for Moldova align with a previous economic evaluation for populations across South Africa, Belarus, and Uzbekistan [28]. Like Moldova, Belarus has a high proportion of RR-TB among individuals newly diagnosed with TB [2], but we do not know whether the joint distribution of resistance to other important drugs would differ between Belarus and Moldova. Although South Africa and Uzbekistan have a lower prevalence of resistance to many drugs, we found that 6 months of BPaLM remained cost-effective when the proportion of patients with FQ-R was varied across the wide range of 0% to 40% (compared to Moldova at 28%). Our analysis builds on the aforementioned cost-effectiveness analysis by explicitly modeling the acquisition of drug resistance, with the initial cohort resistance profile informed by genetic sequencing data from Moldova. We also investigated the potential consequences of a larger

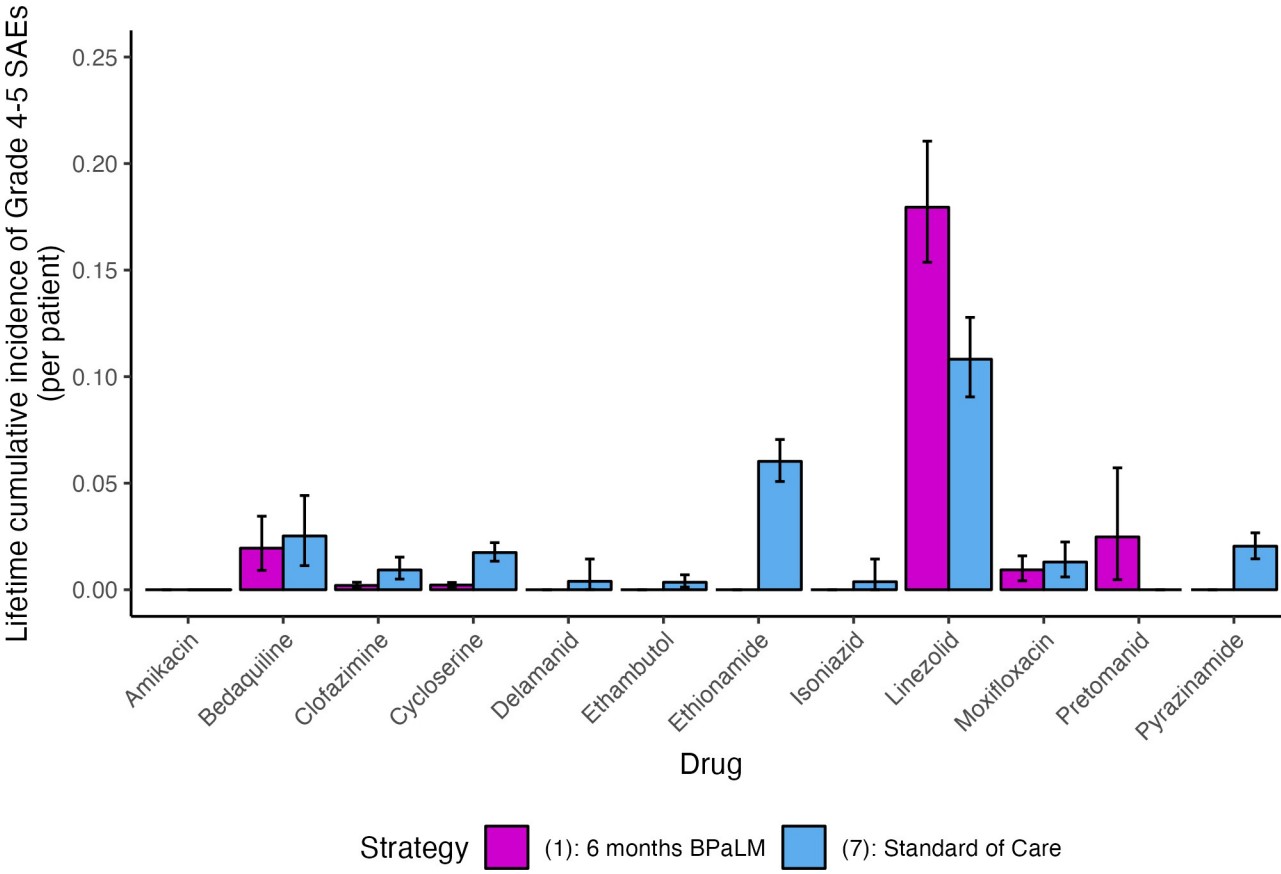

**Fig 4. Cumulative incidence of grade 4–5 SAEs under 6 months BPaLM and standard of care.** The mean cumulative incidence of grade 4–5 SAEs ever experienced to each of 12 anti-TB drugs is shown for Strategy (1) (6 months BPaLM, DST upfront, repeat DST every 4 months, BPaLC if Mfx stopped) as compared to Strategy (7) (standard of care 9- to 18-month regimens based on results of upfront DST, repeat DST every 4 months). Estimates are provided per individual, averaged over the entire cohort initiating treatment. The mean estimate is shown by the bar, with 95% UIs represented as error bars. BPaLM, bedaquiline, pretomanid, linezolid, moxifloxacin; SAE, grade 4–5 severe adverse event; TB, tuberculosis; UI, uncertainty interval.

number of policy implementation scenarios, including the frequency of DST, and whether patients having to stop Mfx under BPaLM should continue on BPaL alone or continue on the alternative four-drug regimen BPaLC.

When modeling the comparative effectiveness of 6 months BPaLM against the standard of care, we assumed that the hazard rate ratio for true cure was reasonably approximated by that for sputum culture conversion in the TB-PRACTECAL trial [8]. Even if the comparative effectiveness for true cure is not the same as on culture conversion, we found that 6-months BPaLM remained the cost-effective strategy when the HRR (point estimate: 1.59) was varied over a wide range. We chose not to build our model around the trial's primary effect measure, which was a composite outcome combining treatment failure, discontinuation, LTFU, death, and recurrence. While each is clinically meaningful in its own right, the impacts on long-term measures of health such as QALYs may differ substantially between each outcome included. In TB-PRACTECAL for example, we note that the biggest component of the reduction in unfavorable outcomes under 6 months of BPaLM was conveyed not by a reduction in deaths but by a decrease in discontinuation (specifically, discontinuations resulting from adverse events and withdrawal of consent) [8]. In fact, our model results illustrate exactly this discrepancy between short-term composite and long-term QALY outcomes: While we estimated that 6

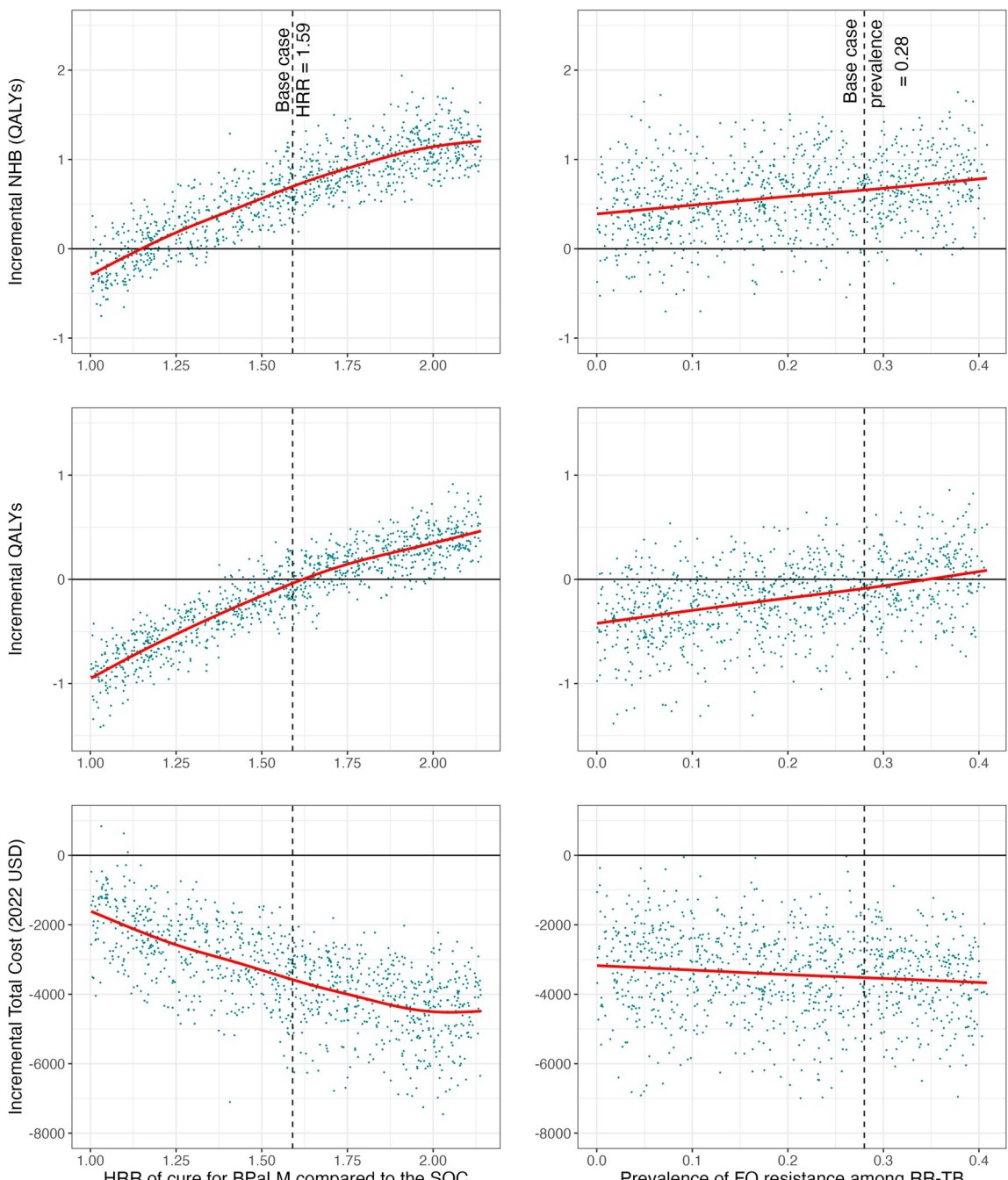

**Fig 5. Sensitivity analyses varying relative effectiveness of BPaLM and cohort prevalence of FQ-R.** One-way sensitivity analyses exploring the implications of key model parameters on the incremental benefits and costs of Strategy (1) (6 months BPaLM, DST upfront, repeat DST every 4 months, BPaLC if Mfx stopped) as compared to Strategy (7) (standard of care 9- to 18-month regimens based on results of upfront DST, repeat DST every 4 months). We chose to compare these 2 strategies as they were the best-performing BPaLM-based and standard of care-based strategies, respectively. In the left column, the HRR of cure for the BPaLM regimen compared to the standard of care was varied. In the right column, we varied the starting prevalence of FQ-R in the cohort (i.e., among all RR-TB). Each of the parameters was varied deterministically in the respective sensitivity analysis, with all other model parameters drawn as in the probabilistic sensitivity analysis. The outcomes quantified on the y-axis for each row of plots are (top to bottom): incremental NHB (calculated using discounted Total Costs and discounted QALYs at the lower bound WTP), incremental QALYs (undiscounted), and incremental Total Costs (undiscounted). The difference between the modeled outcomes under BPaLM and the standard of care is shown for 1,000 model runs, each an average of 10,000 individual patient simulations. The red line shows the trend as represented by regression of the y-axis variable on the x-axis variable, using a generalized additive model with cubic spline to obtain a restricted maximum likelihood within ggplot2 [58]. The vertical dashed lines mark the base case assumptions for the mean of each of

these model parameters. BPaLM, bedaquiline, pretomanid, linezolid, moxifloxacin; DST, drug susceptibility testing; FQ, fluoroquinolone; HRR, hazard rate ratio; NHB, net health benefit; QALY, quality-adjusted life year; RR-TB, rifampicin-resistant tuberculosis; WTP, willingness-to-pay.

months of BPaLM could reduce the risk of unfavorable outcomes at 72 weeks—in line with TB-PRACTECAL [8]—our model showed that 6 months of BPaLM was associated with a non-significant reduction in QALYs when compared to the standard of care. This discrepancy is also partly due to treatment duration; standard of care regimens are administered for 9 to 18 months (i.e., 39 to 78 weeks), and so any treatment impacts accruing toward the end of treatment (beyond 72 weeks) were not captured in the unfavorable outcome endpoint, while they were captured in QALYs which had a lifetime horizon.

While both regimens performed best at lower levels of resistance, sensitivity analyses showed that 6 months of BPaLM may result in a reduction in total QALYs as compared to the standard of care at lower levels of initial FQ-R, or if the BPaLM regimen had a lower comparative effectiveness than estimated by culture conversion in the TB-PRACTECAL trial, even while it provides overall value for money. Although policymakers may be uncomfortable adopting interventions that could reduce health on average, this difference was not statistically significant. Adopting the new regimen would likely bring substantial benefits in the form of reduced regimen duration and by freeing up funding to spend on other health interventions.

We estimated that 6 months of BPaLM improved or resulted in no change to the duration of disease with resistant strains of *M. tuberculosis* as well as the cumulative incidence of resistance for all anti-TB drugs investigated, except pretomanid and delamanid. Both the duration and cumulative incidence measures were influenced by the starting profile of resistance as informed by the WGS data, the rate of acquisition of new resistance to each drug under each modeled drug regimen, and the monthly rate of cure. Changes in the rate of acquisition of resistance are important for individuals undergoing treatment today (some of which is captured in the QALYs estimated under each strategy) but preventing new second-line resistance is also important for the health outcomes of those living with RR-TB in the future.

This analysis had several limitations. The Moldovan genomic data used to characterize the resistance profile in the modeled population were from culture positive sputum specimens in 2018 to 2019; as such they may not accurately describe current resistance patterns in Moldova or resistance elsewhere, although we hope the sensitivity analysis on the prevalence of FQ-R aids in the generalization of findings. Because the publicly available WGS dataset excluded samples with mixed strains of *M. tuberculosis* (17.4%), it is possible that our findings do not adequately address this subpopulation with mixed infections, although we note that all the remaining model parameters reflect the real-world health outcomes and costs of a mix of mono- and mixed-strain infections. Furthermore, we assumed that the true resistance profile was perfectly predicted by the presence or absence of mutations conferring resistance in this data: While the sensitivity and specificity of genomic sequencing is very high for detecting resistance in rifampicin, isoniazid, and ethambutol, the performance is less favorable for moxifloxacin, amikacin, and ethionamide [63].

There are also limitations pertaining to the simulation of health and cost outcomes. The hazard rate ratio for cure was based on the outcome of sputum culture conversion from TB-PRACTECAL; while culture conversion is indeed a prognostic marker in TB [64], it is not a perfect substitute to quantify the rate of true cure, which is unobservable. Further, real-world outcomes with 6 months of BPaLM are likely to be less favorable than in the high-fidelity environment of a randomized controlled trial—for example, there may have been a higher frequency of follow-up visits in the trial than what may be possible in practice—and the status

quo may differ between settings. We did not explicitly model the differences in adherence that may exist between regimens, and we made the simplifying assumption that increasing the number of effective drugs increases the monthly rate of cure and reduces the rate of acquiring resistance. This was based on a previously applied approach [7] and is likely to hold qualitatively, but we did not account for the all the differences that may exist between specific drugs, and the interactions between them. For example, the effectiveness of BPaLC versus BPaL may not be the same as the effectiveness of BPaLM versus BPaL, yet—SAEs aside—the modeling approach was agnostic to this, conditional on the number of "effective" drugs in the regimen. For parsimony, we did not explicitly model changes in smear status. For individuals no longer receiving treatment, we adopted a mortality rate estimate for smear–positive TB, which may overestimate mortality specifically for those who appear to have completed treatment successfully but not truly cured; this would likely bias the results against shorter, 6-month BPaLM strategies. While the relationship between HIV and RR-TB treatment outcomes is neither straight-forward nor consistent [39,65], we did not model HIV status at the individual level and as such we were unable to comment specifically on health outcomes for those with TB-HIV coinfection. Although the probability of a grade 4–5 SAE was modeled separately for each drug, we did not incorporate variation in the duration and consequences of each type of SAE. Finally, we did not account for the secondary impacts resulting from onward transmission of RR-TB, and our results may therefore not capture the full cost-effectiveness implications of each modeled strategy. To account for this explicitly, it would be necessary to model the transmission dynamics of *M. tuberculosis*. Instead, we estimated the cumulative incidence and duration of resistance as surrogates for the long-term health outcomes they may affect, insofar as lower incidence and fewer months of resistant disease might each result in less transmission of resistant strains.

This study was conducted in the setting of Moldova, a country with a high proportion of RR-TB with resistance to second-line drugs. By conducting a sensitivity analysis on the proportion with FQ-R, we aimed to aid the generalization of findings to other settings. Many of the health-related model parameters are also generalizable beyond Moldova: TB outcomes under the standard of care were informed by multinational meta-analyses and the estimate for comparative effectiveness was from a multinational trial (S1 Table). However, many of the cost parameters were from Moldova and Georgia (GDP per capita of $5,563 and $6,628 in 2022, respectively) [66], and so there are likely limitations in the generalization of incremental costs of 6 months BPaLM compared to the standard of care, especially to countries with very different income levels.

To optimize clinical care for RR-TB, decision makers must take into account important health and economic consequences for affected individuals as well as society at large. In this study, we estimated favorable cost-effectiveness for the 6-month BPaLM regimen in settings with a high burden of drug resistance, conditional on BPaLC being used in the event of moxifloxacin being contraindicated, rather than BPaL alone. The schedule of second-line DST did not appear to affect health outcomes or costs to a great degree across the finite number of DST schedules we explored, and further analyses may be warranted to explore the optimal testing frequency in Moldova and other settings—especially where second-line DST capacity is limited or unavailable [67]—and to explore additional technologies beyond MGIT for identifying resistance to fluoroquinolones and injectables. The forthcoming results of the endTB trial [9–13] will expand the evidence base for shorter regimens, and while that trial investigated 9- as opposed to the 6-month regimens we investigated, this still represents a substantial shortening compared to many standard of care regimens. The growing body of both empirical and modeling literature may also highlight the elements of treating RR-TB—including the choice of drugs, duration of regimen, and frequency and modality of DST—which overall provide the

best treatment strategy, for each patient's specific needs. Clinical and health policy decisions alike would continue to be enhanced by collective efforts to strengthen the evidence base in the ways most likely to optimize care, with data of sufficient quantity and quality to characterize long-term health outcomes across multiple settings.

## Supporting information

**S1 Checklist. CHEERS 2022 Checklist.**
(PDF)

**S1 Appendix. Additional detail on microsimulation model.** Here, we provide an enhanced level of detail on some of the model structure. Specific elements reference sources from the literature [43,44,48,68–75].
(PDF)

**S2 Appendix. Additional detail on calculated model parameters.** Here, we provide notation for how specific model parameters were derived from published sources [39,76].
(PDF)

**S1 Table. Model input parameters, complete set.** CDF, cumulative distribution function; CI, confidence interval; CrI, credibility interval; CPI, Consumer Price Index; DST, drug susceptibility testing; LJ, Lowenstein–Jensen; MDL, Moldovan Leu; GDP, Gross Domestic Product; GEL, Georgian Lei; LTFU, lost to follow up; *M. tb.*, *Mycobacterium tuberculosis*; NHB, net health benefit; QALY, quality-adjusted life year; SAE, severe adverse event; SEM, standard error of the mean; SMR, standardized mortality ratio; UI, uncertainty interval; USD, United States dollars; WTP, willingness-to-pay. *Denotes a parameter where there was no readily available measure of dispersion. For these parameters, we assumed a standard deviation equal to one third of the mean. Parameter details are accompanied by citations from the literature [8,14,20,36–42,45,47,66,68,76–92].
(PDF)

**S2 Table. Probability of loss to follow up by month of treatment.** LTFU, lost to follow up. LTFU data from Walker and colleagues [78]. *The values in the rightmost column are used as the model inputs. Compared to the fourth column, we rounded down the values from month 21 onwards such that the probability of LTFU is zero thenceforth.
(PDF)

**S3 Table. Duration and cumulative incidence of resistance to key drugs.** BPaLM, bedaquiline, pretomanid, linezolid, moxifloxacin; SOC, standard of care. The entire cohort had RR-TB, and so the duration with rifampicin resistance is equivalent to the duration with active RR-TB, and the cumulative incidence of rifampicin resistance is not applicable. Some drugs were used very sparingly, if ever, under one or both strategies (e.g., amikacin, ethambutol, ethionamide, isoniazid, and pyrazinamide); as such the cumulative incidence may be very low for these drugs under one or both strategies.
(PDF)

**S4 Table. Life years achieved under each RR-TB treatment strategy.** BPaL, bedaquiline, pretomanid, linezolid; BPaLC, bedaquiline, pretomanid, linezolid, clofazimine; BPaLM, bedaquiline, pretomanid, linezolid, moxifloxacin; UI, uncertainty interval. Strategies are listed in the same order as Table 3. Mean values are shown with accompanying 95% UIs in parentheses.
(PDF)

**S5 Table. Comparing outcomes at 6 months, 12 months, and 72 weeks from treatment initiation.** BPaLC, bedaquiline, pretomanid, linezolid, clofazimine; BPaLM, bedaquiline, pretomanid, linezolid, moxifloxacin; LYs, life years; Mfx, moxifloxacin; p.p., percentage points; QALYs, quality-adjusted life years; UI, uncertainty interval; WHO, World Health Organization. The following health outcomes are shown: a composite "Unfavorable outcome" closely aligned to the composite trial endpoint in TB-PRACTECAL, true cure, and quality-adjusted life expectancy. Results are shown separately over 3 model-run time horizons: 6 months, 72 weeks (in line with the endpoint in TB-PRACTECAL), and lifetime (in line with the primary outcomes in our analysis).
(PDF)

**S6 Table. Budget impact over 5 years of implementing 6 months of BPaLM in Moldova.** TB, tuberculosis. The budget impact was estimated for 6 months BPaLM (Strategy (1)) as compared to standard of care (Strategy (7)).
(PDF)

**S1 Fig. Schematic of the initial workup phase for the standard of care.** Both standard of care strategies (Strategy 7 and Strategy 8) are modeled on the recommended workup and regimen selection in the 2020 WHO guidelines on the treatment of drug-resistant tuberculosis [2]. We assumed that DST results (by MGIT) are available in 2 weeks. *While we include the BPaL regimen as per the guidelines, no patients actually met the criteria to receive it under the standard of care (Strategies 7 and 8) in our model (i.e., in all model simulations, it is possible to adopt a WHO longer regimen). BPaL, bedaquiline, pretomanid, linezolid; DST, drug susceptibility test; FQ, fluoroquinolone; MGIT, mycobacterial growth indicator tube; WHO, World Health Organization.
(PDF)

**S2 Fig. *M. tb.* genomic sequencing data exclusion criteria.** *Specimens demonstrating polyclonal infections were already excluded (*n* = 386), leaving a full dataset of 1,834 *M. tb* isolates. Exclusions made to the genomic sequencing drug susceptibility testing dataset are shown along with the number of observations. This dataset of pretreatment isolates is described elsewhere [30,35]. The presence of a mutation conferring resistance to rifampicin was assumed to convey full resistance and vice versa. The dataset with exclusion criteria applied is available at https://github.com/lyndonpjames/BPaLM_Moldova/blob/main/tbl_WGS_allRR.csv while original publicly available datasets can be found at https://www.ncbi.nlm.nih.gov/biosample?Db=biosample&DbFrom=bioproject&Cmd=Link&LinkName=bioproject_biosample&LinkReadableName=BioSample&ordinalpos=1&IdsFromResult=736718 [93] and https://www.ncbi.nlm.nih.gov/pmc/articles/PMC8903246/bin/pmed.1003933.s002.csv [30]. TB, tuberculosis.
(PDF)

**S3 Fig. The rate of acquiring drug resistance.** The modeled point estimate for the monthly rate that an individual's strain of *M. tuberculosis* will acquire resistance to each effective drug it is exposed to is plotted, conditional on that individual beginning the month with *n* effective drugs in the regimen (x-axis). Estimates for 1, 3, and 4 effective drugs were obtained from the literature. The estimate for 2 drugs was calculated, assuming an additive risk (i.e., the increase in risk for 2 effective drugs compared to 3 is the same as the increase in risk for 3 effective drugs compared to 4). See also S1 Table.
(PDF)

**S4 Fig. Markov state-transition diagram.** Transitions between states can occur as shown by the arrows. Though not receiving treatment, individuals in the "Active TB, no longer receiving treatment" state are subject to a low rate of self-cure, and so may still transition to the "Cured post-treatment" state. Asterisks (*) highlight the major mechanisms through which the choice of treatment intervention affects outcomes. LTFU, lost to follow-up; TB, tuberculosis. Images within this figure were obtained as icons from Microsoft with no license or terms of use: https://support.microsoft.com/en-us/office/insert-icons-in-microsoft-365-e2459f17-3996-4795-996e-b9a13486fa79?ui=en-us&rs=en-us&ad=us.
(PDF)

**S5 Fig. Cohort prevalence of *M. tb*. resistance to key drugs at treatment initiation.** The proportion of the cohort with primary resistance to each drug is plotted, as described by *M. tuberculosis* whole genomic sequencing data from Moldova [30,35]. All those observations with rifampicin susceptibility were excluded, as per S2 Fig. *There was no resistance data for pretomanid; resistance was assumed to be at the same level as for delamanid.
(PDF)

**S6 Fig. WHO-based definitions for end-of-treatment outcomes.** Schematic showing the definitions for end-of-treatment outcomes used by the WHO (A), and this model (B). The constituents of each of the major end of treatment outcome categories are shown, as applied to all RR-TB including MDR-TB and XDR-TB. Differences between the WHO definitions and those used in this model are highlighted by the gray hashed boxes. The WHO definitions are not necessarily mutually exclusive; we assumed that the classification takes place according to the tree structure in (A), and implemented the aligned structure in (B) for tractability given the model mechanisms. For example, an individual who failed treatment and then died would be recorded as a death, because the branch involving death is closer to the root of the tree.
(PDF)

**S7 Fig. Incremental cost-effectiveness plane for the leading 6-month BPaLM strategy vs. the leading SOC strategy.** The incremental cost-effectiveness plane compares the incremental discounted total QALYs and incremental discounted total costs for Strategy (1) as compared to a reference of Strategy (7). Each light pink point represents 1 iteration of the second-order Monte Carlo simulation, itself an average of 10,000 individual patient simulations. The purple diamond is the mean of the 1,000 second-order Monte Carlo simulations, corresponding to the point estimates in Table 3. The blue dot represents the standard of care (Strategy (7)), which is the reference point. BPaL, bedaquiline, pretomanid, linezolid; BPaLC, bedaquiline, pretomanid, linezolid, clofazimine; BPaLM, bedaquiline, pretomanid, linezolid, moxifloxacin; FQ-R, fluoroquinolone-resistant; FQ-S, fluoroquinolone-susceptible; Mfx, moxifloxacin; QALY, quality-adjusted life year; USD, United States dollars; WTP, willingness-to-pay.
(PDF)

**S8 Fig. Sensitivity analyses varying relative effectiveness of BPaLM and cohort prevalence of fluoroquinolone resistance, outcome of life years.** These one-way sensitivity analyses tested the impact of key model parameter assumptions on the incremental life years experienced under the Strategy (1) (6 months BPaLM, DST upfront, repeat DST every 4 months, BPaLC if Mfx stopped) as compared to Strategy (7) (standard of care 9- to 18-month regimens based on results of upfront DST, repeat DST every 4 months). We chose to compare these 2 strategies as they were the best-performing BPaLM-based and standard of care-based strategies, respectively. Each of the parameters is varied deterministically in the respective sensitivity analysis, with all other model parameters drawn as in the probabilistic sensitivity analysis. In the left column, the HRR of cure for the BPaLM regimen compared to the standard of care is

varied. In the right column, we vary the starting prevalence of fluoroquinolone resistance in the cohort. Each of 1,000 model runs is shown in each plot, itself an average of 10,000 individual patient simulations. The red line shows the trend as represented by regression of the y-axis variable on the x-axis variable, using a generalized additive model with cubic spline to obtain a restricted maximum likelihood within ggplot2 [58]. The vertical dashed lines mark the base case assumption for the mean of each of these model parameters. FQR, fluoroquinolone resistance; HRR, hazard rate ratio; LY, life year; RR-TB, rifampicin-resistant tuberculosis. (PDF)

**S9 Fig. Validating modeled end-of-treatment outcomes against data reported to WHO.** The proportions recorded for each EOT outcome are shown for WHO RR-TB data 2010–2019 for Moldova (left of the vertical dashed line) [94] and the modeled cohort outcomes (right of the vertical dashed line), where we assume that all EOT outcome categories are mutually exclusive, and that death during treatment or LTFU take precedence over a preceding treatment failure. Standard of care refers to modeled Strategy 7, and 6 months BPaLM refers to modeled Strategy 1. The number of observations per year in the WHO TB outcomes data for all MDR/ RR-TB is in the range (559, 996). BPaLM, bedaquiline, pretomanid, linezolid, moxifloxacin; EOT, end-of-treatment; LTFU, lost to follow up; RR-TB, rifampicin-resistant tuberculosis; WHO, World Health Organization. (PDF)

## Acknowledgments

For helpful feedback on research-in-progress presentations, LPJ would like to thank current and former students, postdocs, and faculty affiliated with: the Center for Health Decision Science, the PhD Program in Health Policy and the Center for AIDS Research, both at Harvard University, Cambridge, MA, USA; and the Decision Science Methods Group at Erasmus University, Rotterdam, the Netherlands.

## Author Contributions

**Conceptualization:** Lyndon P. James, Jennifer Furin, Reza Yaesoubi, Ted Cohen, Nicolas A. Menzies.

**Data curation:** Lyndon P. James, Sedona Sweeney, Jennifer Furin, Nicolas A. Menzies.

**Formal analysis:** Lyndon P. James.

**Funding acquisition:** Lyndon P. James, Nicolas A. Menzies.

**Investigation:** Lyndon P. James, Sedona Sweeney, Jennifer Furin, Molly F. Franke, Reza Yaesoubi, Dumitru Chesov, Nelly Ciobanu, Alexandru Codreanu, Valeriu Crudu, Ted Cohen, Nicolas A. Menzies.

**Methodology:** Lyndon P. James, Sedona Sweeney, Reza Yaesoubi, Ted Cohen, Nicolas A. Menzies.

**Project administration:** Lyndon P. James, Nicolas A. Menzies.

**Resources:** Valeriu Crudu, Nicolas A. Menzies.

**Software:** Lyndon P. James, Fayette Klaassen, Nicolas A. Menzies.

**Supervision:** Molly F. Franke, Ted Cohen, Nicolas A. Menzies.

**Validation:** Lyndon P. James, Molly F. Franke, Ted Cohen, Nicolas A. Menzies.

**Visualization:** Lyndon P. James, Fayette Klaassen.

**Writing – original draft:** Lyndon P. James, Nicolas A. Menzies.

**Writing – review & editing:** Lyndon P. James, Fayette Klaassen, Sedona Sweeney, Jennifer Furin, Molly F. Franke, Reza Yaesoubi, Dumitru Chesov, Nelly Ciobanu, Alexandru Codreanu, Valeriu Crudu, Ted Cohen, Nicolas A. Menzies.

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
