## [Editor Report · Decision Letter 0]

2 Aug 2023

Dear Dr James, 

Thank you for submitting your manuscript entitled "Impact and cost-effectiveness of the 6-month BPaLM regimen for rifampicin-resistant tuberculosis: a mathematical modeling analysis" for consideration by PLOS Medicine.

Your manuscript has now been evaluated by the PLOS Medicine editorial staff as well as by an academic editor with relevant expertise and I am writing to let you know that we would like to send your submission out for external peer review.

Please re-submit your manuscript within two working days, i.e. by Aug 04 2023 11:59PM.

Kind regards,

Katrien Janin

Senior Editor

PLOS Medicine

---

## [Decision Letter · Decision Letter 1]

23 Oct 2023

Dear Dr. James,

Thank you very much for submitting your manuscript "Impact and cost-effectiveness of the 6-month BPaLM regimen for rifampicin-resistant tuberculosis: a mathematical modeling analysis" (PMEDICINE-D-23-02158R1) for consideration at PLOS Medicine. 

[LINK]

In light of these reviews, I am afraid that we will not be able to accept the manuscript for publication in the journal in its current form, but we would like to consider a revised version that addresses the reviewers' and editors' comments. Obviously we cannot make any decision about publication until we have seen the revised manuscript and your response, and we plan to seek re-review by one or more of the reviewers. 

We expect to receive your revised manuscript by Nov 13 2023 11:59PM. Please email us (plosmedicine@plos.org) if you have any questions or concerns.

We look forward to receiving your revised manuscript. 

Sincerely,

Katrien Janin, PhD

PLOS Medicine

plosmedicine.org

GENERAL: Please provide 95% UIs and p values for all results were appropriate (including the abstract), check and amend throughout. We suggest reporting statistical information in the following format: ‘x’; (95% UI [‘y’,’ z’] p value) For p values, please report as p<0.001 and where higher as 'p=0.002'. Please add the p value statistical method used to your method section.

For in-text reference, citation reference numbers are placed within square parentheses and these should precede punctuation. Please amend throughout. 

STUDY DESIGN:

MODELLING STUDIES 

(For the Microsimulation Model)

Of all authors who submit a modelling study we ask for inclusion of specific items, derived from Geoffrey P Garnett, Simon Cousens, Timothy B Hallett, Richard Steketee, Neff Walker. Mathematical models in the evaluation of health programmes. (2011) Lancet DOI:10.1016/S0140-6736(10)61505-X.

Please ensure all the items listed below are included with your manuscript. Please review the list below and confirm/revise as necessary: 

(i) Please provide a diagram that shows the model structure, including how the disease natural history is represented, the process and determinants of disease acquisition, and how the putative intervention could affect the system.

(ii) Please provide a complete list of model parameters, including clear and precise descriptions of each parameter, together with the values or ranges for each, with justification or the primary source cited, and important caveats about the use of these values noted.

(iii) Please provide a clear statement about how the model was fitted to the data.

(iv) For uncertainty analyses, please state the sources of uncertainties quantified and not quantified.

(v) Please provide sensitivity analyses to identify which parameter values are most important in the model. Uncertainty estimates seek to derive a range of credible results on the basis of an exploration of the range of reasonable parameter values. The choice of method should be presented and justified.

(vi) Please discuss the scientific rationale for this choice of model structure and identify points where this choice could influence conclusions drawn. Please also describe the strength of the scientific basis underlying the key model assumptions.

We thank you for including the CHEERS checklist to cover the cost effectiveness part of the analyses. 

DATA AVAILABILITY: For studies in which a model is central to the manuscript's findings, as is the case here, authors are responsible for providing the source code needed to replicate the study's findings in a repository (such as GitHub, SourceForge or Bitbucket) or a cloud computing service (such as Code Ocean). Protection of authors’ intellectual property will not be cause for exception. Please explain in the manuscript’s Data Availability Statement how readers can access the shared code. 

ABSTRACT: In Abstract Methods and Findings: As above, please quantify the main results (with 95% UIs and p values).

In the last sentence of the Abstract Methods and Findings section, please describe the main limitation(s) of the study's methodology.

AUTHORS SUMMARY:

Ideally each sub-heading should contain 2-3 single sentence, concise bullet points containing the most salient points from your study.

In the final bullet point of ‘What Do These Findings Mean?’ Please include the main limitations of the study in non-technical language.

ACKNOWLEDGMENTS/ DECLARATIONS

Please remove all statements apart from acknowledgements, author contributions and abbreviations from the end of the main manuscript and include these only in the relevant parts of the manuscript submission form. Funding, competing interest, and data availability will be compiled as metadata.

REFERENCES:

Please use the "Vancouver" style for reference formatting, and see our website for other reference guidelines https://journals.plos.org/plosmedicine/s/submission-guidelines#loc-references

As mentioned above, for in-text reference, citations are placed within square parentheses and should precede punctuation. Please amend throughout. 

Please also ensure that any references to online-only sources includes a date of accession. Please use [date accessed: ] format instead of ‘cited’. Thank you. 

FIGURES and TABLES;

Please ensure you re-define abbreviations in the legends of figure and tables (e.g Figure 1, 2, 3, etc - Table 2). 

Comments from the reviewers:

Reviewer #1: Thank you for the opportunity to review the manuscript by James and colleagues, addressing the cost-effectiveness of treatment and drug susceptibility testing strategies for persons with rifampicin-resistant TB in Moldova. The study findings suggest that adoption of the 6-month BPaLM regimen will be cost-effective compared to the previous standard regimens: in the long term, there are likely to be cost savings, with relatively similar quality-adjusted survival.

Major Comments:

1. There are several strengths in this analysis, including its use of Moldovan data with respect to evolving drug susceptibilities. The modeling approach and assumptions are reasonable and generally well described, with suitable sensitivity analyses. Many of the potential limitations are appropriately acknowledged.

2. That said, there are some areas for clarification and comment. The first relates to the analysis of quality-adjusted survival. As this is the major health outcome considered, the assumptions regarding QALY weights, and their justification, need to be included in the main text and tables. In the current version of the manuscript, this is mentioned in a cursory manner (p. 7, lines 66-68) with an incorrect reference. There is more information available in the supplement, but key elements of this including suitable explanation and references should be moved to the main text.

3. Also related to the QALY outputs, in sensitivity analysis the leading BPaLM strategy was led to an expected decrement of 0.55 QALYs when the hazard rate ratio for cure with that regimen was 1.2 relative to the standard RR-TB regimen. This is described as "small, non-significant" [p. 15. Line 33]. I suspect many readers would disagree and I would suggest that this description be dropped. Perhaps more importantly, the reasons for this non-trivial decrement were not clear—it occurs despite improved cure rates with BPaLM [i.e. with HRR remaining at 1.2]. It did not seem to be due to an increased frequency of severe adverse events. Even in the base case analysis, BPaLM was associated with a slight decrement in quality-adjusted survival, despite a more marked assumed improvement in cure rates. Why is this?

4. Some of the key model input parameters (table S1) should be moved to a table accompanying the main text—the most important natural history, treatment-related, QALY, and cost parameters, for example. The text in the main methods section outlining cost assumptions should point to the appropriate table(s) [p. 8, lines 85-93].

5. The accounting for direct non-medical and indirect costs of RR-TB and its treatment (items 64 and 65) in table S1 is a bit confusing. The point estimates are listed as $2,044 and $2,301 respectively. If I have understood correctly, these are based on an 18-month treatment episode, and then converted to monthly estimates. So for individuals who receive 6 months of treatment, are the point estimates 1/3 of those amounts? And for 9 months, are they half?

6. The identification of dominated vs. non-dominated strategies is based on point estimates for QALYs that fall very close together. The same holds true for cost estimates for the various antibiotic replacement and susceptibility testing strategies, when the initial treatment regimen is held constant. There is huge overlap between the various uncertainty ranges. This at least deserves comment. It is certainly most efficient, given limited space, to focus on what appear to be the "leading" BPaLM and "conventional treatment" strategies, but differences between these and the "dominated" strategies should not be overstated.

7. For this reason, it would be appropriate to show standard cost-effectiveness planes, showing the various point estimates for incremental costs and QALYs from probabilistic analysis, comparing strategies 1 and 7. Some of this information is of course implicit in figure 5, but it should be shown explicitly in a separate figure.

8. The authors appropriately acknowledge that they have attributed hazard rate ratios for cure for BPalM based on those observed for sputum culture conversion in the TB-PRACTECAL study. They recognize that sputum culture conversion may be an imperfect surrogate for cure. With that in mind, it would be useful to extend the sensitivity analysis around the HRR for cure down to a value of 1, i.e. BPaLM exactly equivalent to standard treatment with respect to cure rate. This would further address uncertainty related to extrapolation of cure rates from sputum conversion rates.

9. Figure 4 could potentially be moved to the supplement. 

10. The modeling approach used here does not account for secondary transmission of RR-TB, and its downstream impacts. Given the scope and nature of the research question, this is not a major limitation in my view, but it should be acknowledged explicitly in the discussion section.

Minor Comments:

1. Page 11, line 21 and figure 1B: The probabilities shown are of the strategy in question being the MOST cost-effective, compared to the others—not just the probability of being cost-effective according to the willingness-to-pay threshold shown. Please change the description in the text, figure legend and labels accordingly.

2. Table 1: would be clearer to state "Every 4 months" or "Every 1 month" under "Routine frequency of subsequent DST."

Reviewer #2: In addressing the complexities of treating Rifampicin-Resistant Tuberculosis (RR-TB), the WHO's 2022 guidelines introduced a shorter, 6-month BPaLM regimen, contrasting with the earlier 2020 guidelines which recommended longer treatments spanning 9-18 months. This study employed a detailed mathematical model to estimate the long-term health outcomes and costs of these strategies in Moldova, a country with a high TB drug resistance burden. Through the utilization of genomic sequencing and a Markov microsimulation model, the research evaluated regimen effectiveness, potential severe adverse events, and resistance acquisition. The results indicated that the shorter

---

## [Decision Letter · Decision Letter 2]

26 Feb 2024

Dear Dr. James,

Thank you very much for re-submitting your manuscript "Impact and cost-effectiveness of the 6-month BPaLM regimen for rifampicin-resistant tuberculosis: a mathematical modeling analysis" (PMEDICINE-D-23-02158R2) for review by PLOS Medicine.

I have discussed the paper with my colleagues and the academic editor and it was also seen again by the reviewers. I am pleased to say that provided the remaining editorial and production issues are dealt with we are planning to accept the paper for publication in the journal.

[LINK]

We expect to receive your revised manuscript within 10 days. Please email us (plosmedicine@plos.org) if you have any questions or concerns.

We look forward to receiving the revised manuscript by Mar 06 2024 11:59PM.   

Sincerely,

Katrien Janin, PhD

Senior Editor 

PLOS Medicine

plosmedicine.org

Comments from the academic editor 

To ensure the readers are aware that these data should only be generalised to other settings with high RR background rates and similar income level and that further impact and CE research is needed. This can be done by adding more detail to the study setting in:

(1) the title "Impact and cost-effectiveness of the 6-month BpALM regimen for rifampicin-resistant tuberculosis IN MOLDOVA: a..."

(2) the abstract conclusion: currently says "particularly in settings where current long-course regimens are challenging to implement and afford. Further research may be warranted to explore the suitability of shorter RR-TB regimens in specific national settings"

but not sure whether long course regimen implementation and affordability challenges are the generalisable factors. Would rephrase to

"particularly in settings with a high burden of drug-resistant TB. Further research may be warranted to explore the impact and CE of shorter RR-TB regimens across settings with varied drug resistant TB burden and national income levels."

Comments from Reviewers:

Reviewer #1: Thank you for the opportunity to re-review this very well-written manuscript. And thank you to the authors for their extensive and careful revisions. I have identified three areas where I believe further revision/clarification would be helpful.

1. I think the main manuscript would be further strengthened by the addition of a few key QALY and cost parameters to table 1. I realize that there are many parameters and assumptions within each category, but a few key or high-level examples would be helpful to readers. For example, the QALY decrement associated with an episode of RR-TB disease that is successfully treated with one of the six-month regimens [I recognize that this integrates multiple monthly QALY weights, which are detailed in the supplement]. On the cost side, an example could be the "typical" combined drug costs for each of the main regimens modeled. Of course, readers who are particularly interested in detailed aspects of the modeling and the parameters used will refer to the supplementary materials, but placing such items in the main manuscript allows all readers to get some of the context and framing around these parameters.

2. Along similar lines, the authors' description of QALY weights in the main text (p. 16) was not exactly what I meant in my earlier comments—I apologize for any lack of clarity on my part. I was hoping that the authors could provide a clearer picture (and perhaps justification) of the QALY weights they attributed to people undergoing treatment for RR-TB. The added theoretical definition and the discussion of key assumptions underlying the general use of QALYs are interesting for readers, but not essential to the main manuscript, in my view.

3. Perhaps most importantly, I am still struggling to understand the discordance between expected QALYs and favorable outcomes in the primary comparison between the BPaLM [C] and the "standard of care" regimens. I appreciate the authors' clarification that the hazard ratios refer to monthly cure rates, so one must consider that the standard regimens last substantially longer--even if the monthly cure rate is less. But the model outputs also include an overall reduction in unfavorable outcomes with the shorter regimens—as found in the TB-PRACTECAL trial—so why are QALYs lower with the shorter regimens? Again, this does not appear to reflect adverse events. I believe this discordance requires clarification and specific explanation in the text. If there is an explanation which I have missed, I apologize.

A minor point: The Y-axis label in figure 1B should be corrected to read, "Probability of being most-cost-effective."

Reviewer #2: In my opinion, the authors have sufficiently addressed my and other reviewers' questions and concerns and thus the paper is acceptable in its current form. Congratulations to the authors!

Comment from the editorial team

1. Please address the outstanding comments from the academic editor and the reviewer. 

2. Given the study design, please do not use causal language. "Effect" should be used only if causality can be inferred, i.e., for an RCT. Please check and amend through (including the authors’ summary).

3. Data Availability statement. Thank you for agreeing to make your data available. At this time, please provide the direct link to the data file within the repository. Please also add this link to the description of Figure S2, the figure where you describe how you obtained your data set (and the exclusion criteria you applied)

4. Please rephrase sentences that start with a number. (e.g. see line 49 and 51). Please check and amend throughout. 

5. Please refer to people/patients as … people/patients with TB or RR-TB instead of (RR-)TB patients. 

6. General comment - supplementary materials: Please note that supplementary materials are not checked and will be posted as supplied by the authors. Therefore, please double check. Please cite your Supporting Information as outlined here: https://journals.plos.org/plosmedicine/s/supporting-information - Please note you may use almost any description as the item name of your supporting information as long as it contains an "S" and number. For example, “S1 Appendix” and “S2 Appendix,” “S1 Table” and “S2 Table. 

7. Can you check that S5 Figure (S5 Fig) has a main body text call-out? It seems to have been omitted, please double check. I have the same comment for S6 Figure (S6 Fig) and (S8 Fig)

Please do not hesitate to contact me directly at kjanin@plos.org if you have any questions about the above

Best wishes,

Katrien 

[LINK]

---

## [Decision Letter · Decision Letter 3]

10 Apr 2024

Dear Dr James, 

On behalf of my colleagues and the Academic Editor, I am pleased to inform you that we have agreed to publish your manuscript "Impact and cost-effectiveness of the 6-month BPaLM regimen for rifampicin-resistant tuberculosis in Moldova: a mathematical modeling analysis" (PMEDICINE-D-23-02158R3) in PLOS Medicine.

PRESS

Sincerely, 

Katrien G. Janin, PhD 

Senior Editor 

PLOS Medicine